# *Atoh1*-dependent rhombic lip neurons are required for temporal delay between independent respiratory oscillators in embryonic mice

**Srinivasan Tupal[1‡], Wei-Hsiang Huang[2,3†§], Maria Cristina D Picardo[4†], Guang-Yi Ling[1†], Christopher A Del Negro[4], Huda Y Zoghbi[2,3,5], Paul A Gray[1*]**

[1]Department of Anatomy and Neurobiology, Washington University School of Medicine, St Louis, United States; [2]Program in Developmental Biology, Baylor College of Medicine, Houston, United States; [3]Jan and Dan Duncan Neurological Research Institute at Texas Children's Hospital, Baylor College of Medicine, Houston, United States; [4]Department of Applied Science, The College of William and Mary, Williamsburg, United States; [5]Department of Molecular and Human Genetics, Howard Hughes Medical Institute, Baylor College of Medicine, Houston, United States

**\*For correspondence:** pgray@pcg.wustl.edu

†These authors contributed equally to this work

**Present address:** ‡Department of Neurology, University of Virginia, Charlottesville, United States; §Department of Biology, Howard Hughes Medical Institute, Stanford University, Stanford, United States

**Reviewing editor**: Ronald L Calabrese, Emory University, United States

**Abstract** All motor behaviors require precise temporal coordination of different muscle groups. Breathing, for example, involves the sequential activation of numerous muscles hypothesized to be driven by a primary respiratory oscillator, the preBötzinger Complex, and at least one other as-yet unidentified rhythmogenic population. We tested the roles of *Atoh1-*, *Phox2b-*, and *Dbx1*-derived neurons (three groups that have known roles in respiration) in the generation and coordination of respiratory output. We found that *Dbx1*-derived neurons are necessary for all respiratory behaviors, whereas independent but coupled respiratory rhythms persist from at least three different motor pools after eliminating or silencing *Phox2b-* or *Atoh1*-expressing hindbrain neurons. Without *Atoh1* neurons, however, the motor pools become temporally disorganized and coupling between independent respiratory oscillators decreases. We propose *Atoh1* neurons tune the sequential activation of independent oscillators essential for the fine control of different muscles during breathing.

## Introduction

One of the most essential and seemingly simple behaviors in mammals; breathing, requires state-dependent temporal coordination of multiple muscle groups, including the diaphragm and upper airway, to ensure unobstructed airflow (*Feldman et al., 2013*). All motor behaviors, in fact, require the coordinated activity of various muscles acting with temporal precision (*Grillner, 2006*, *2011*), but the neural mechanisms underlying such temporal coordination are only starting to become clear. Identifying how neural networks generate coordinated respiratory behaviors should provide insight into how the brain organizes other complex motor behaviors.

Breathing in mammals encompasses several distinct behaviors, including inspiration, expiration, and sighing (*Ramirez and Viemari, 2005*). Deflection of the diaphragm (inspiration) is the predominant breathing behavior in mammals and requires glutamatergic neurons within the preBötzinger Complex (preBötC) (*Bouvier et al., 2010*; *Gray et al., 2010*; *Gray, 2013*). Expiration is a state-dependent behavior in adults and is normally passive but becomes active under some conditions, including exercise or sleep (*Pagliardini et al., 2012*; *Feldman et al., 2013*). The appearance of rhythmic

**eLife digest** A healthy adult at rest will breathe in and out around 20 times per minute. Each breath requires a complex series of coordinated muscle activity. Inhalation begins with the opening of the airway followed by the contraction of the diaphragm and the intercostal muscles between the ribs, causing the chest cavity to expand. As the lungs increase in volume, the pressure inside them drops and air is drawn in. Relaxation of the diaphragm and intercostal muscles compresses the lungs, causing us to exhale.

Breathing is driven by the brainstem and it cannot be suppressed indefinitely: holding your breath eventually triggers a reflex that forces breathing to resume. The region of the brainstem that controls breathing is called the preBötzinger Complex. However, there is increasing evidence that a second region in the brainstem is also involved. This region, which is called the retrotrapezoid nucleus/parafacial respiratory group, consists of three types of excitatory neurons—Dbx1 neurons, Phox2b neurons, and Atoh1 neurons—but their roles had not been clear. Now, using multiple lines of genetically modified mice, Tupal et al. have teased apart the roles of these three cell types.

These experiments showed that the Dbx1 neurons—which are also found in the preBötzinger Complex—have an essential role in sending the signals from the brain that drive the different muscle activities needed to breathe. The Phox2b neurons modulate breathing based on the level of carbon dioxide in the blood. Atoh1 neurons help control the sequence of respiratory muscle activity during a breath, probably by selectively inhibiting different populations of Dbx1 neurons.

The work of Tupal et al. indicates that distinct populations of neurons within the brainstem independently control two different aspects of breathing: the generation of breathing rhythms, and the coordination of these rhythms. Given that many other physiological processes involve rhythmic activity patterns, this model may help us to understand how the brain generates and controls complex behaviors more generally.

abdominal activity, in vivo, or rhythmic facial cranial (VII) nerve output, in vitro, after opioid-induced inhibition of inspiration, together with data from transection experiments, have provided evidence for an independent second respiratory oscillator adjacent to the VII motor nucleus (*Mellen et al., 2003*; *Janczewski and Feldman, 2006*; *Onimaru et al., 2006, 2009*).

Whereas the vast majority of excitatory neurons in the preBötC are derived from neural progenitors expressing the transcription factor (TF) *Developing Brain Homeobox 1* (*Dbx1*) (*Bouvier et al., 2010*; *Gray et al., 2010*; *Gray, 2013*), the VII region contains three developmentally distinct excitatory neuronal populations that are prime candidates for generating abdominal and VII activity (*Figure 1A*). Although none has yet been conclusively shown to play a role in abdominal or VII rhythm generation (*Feldman et al., 2013*), all three populations express neurokinin 1 receptors (NK1R) and respond to Substance P (SP), and are theorized to couple with preBötC *Dbx1*-derived glutamatergic neurons to generate coordinated respiratory behaviors (*Figure 1B*; *Bouvier et al., 2010*; *Gray et al., 2010*; *Feldman et al., 2013*).

First is a population of glutamatergic retrotrapezoid nucleus (RTN) neurons expressing the TF *Paired-like Homeobox 2b* (*Phox2b*, hereafter called RTN neurons, *Figures 1A and 2A,B*). In adult rodents, these neurons play an important role in chemosensitivity to $CO_2$ (*Stornetta et al., 2006*; *Guyenet and Mulkey, 2010*; *Feldman et al., 2013*). In late embryonic and perinatal rodent reduced preparations, however, RTN neurons are active prior to and independent of those in the preBötC, are rhythmically active out of phase with phrenic motor output, and have been proposed to constitute an independent respiratory oscillator (*Onimaru et al., 2009*; *Thoby-Brisson et al., 2009*). Many of these neurons overlap the functionally defined parafacial respiratory group (pFRG) (*Onimaru and Homma, 2003*; *Onimaru et al., 2009*). RTN neurons transiently express and require the TF *Atonal Homolog 1* (*Atoh1*) post-mitotically for appropriate migration, maturation, and function (*Dubreuil et al., 2009*; *Rose et al., 2009b*). The second RTN/pFRG population consists of *Dbx1*-derived glutamatergic neurons developmentally related to those of the preBötC but whose role is unknown (*Figures 1A and 2C,D*). Loss of *Dbx1* leads to perinatal lethality due, it is assumed, to loss of preBötC and other hindbrain glutamatergic neurons (*Figure 1A,B*; *Gray et al., 2010*).

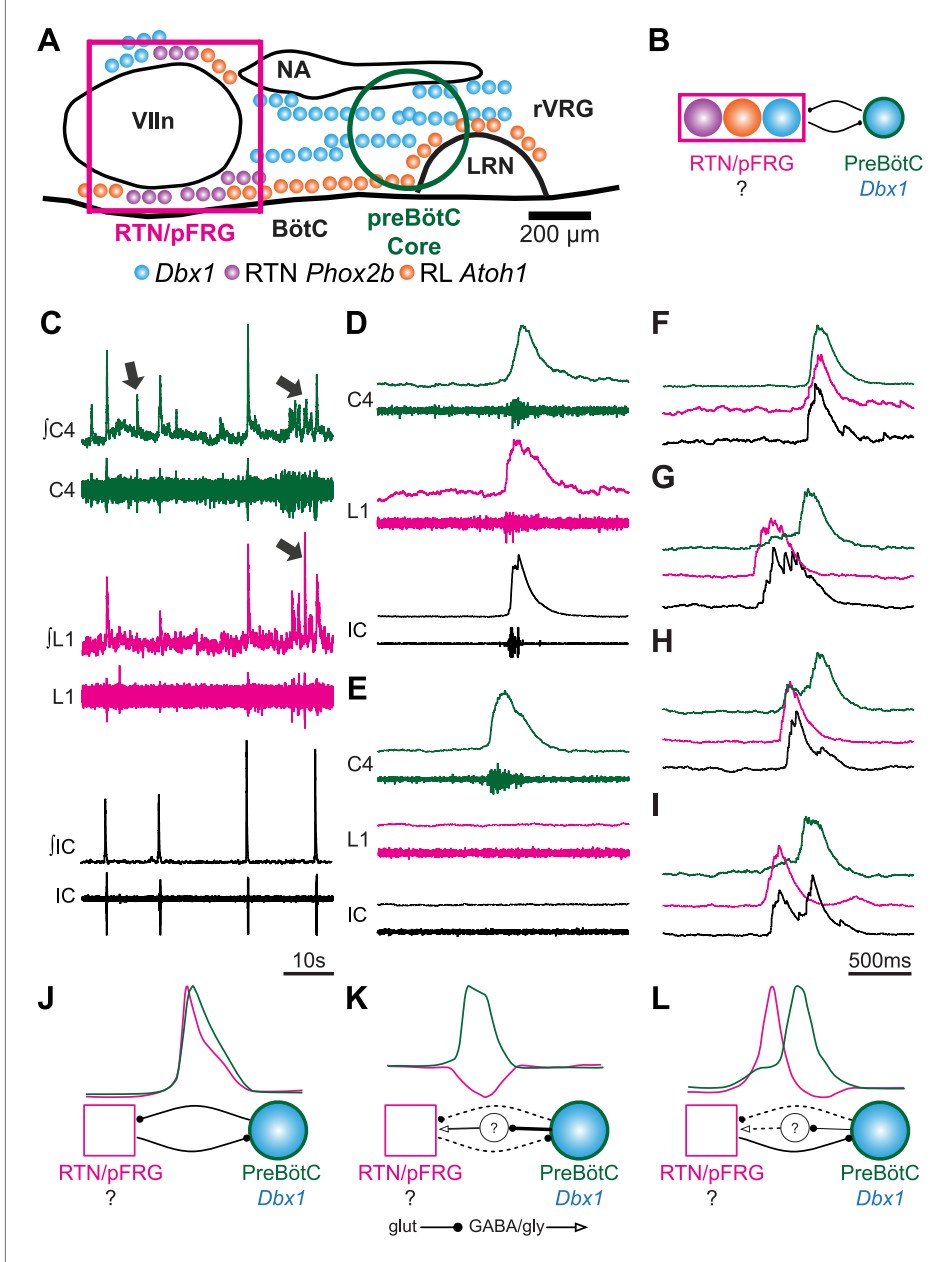

**Figure 1**. Perinatal mouse hindbrain produces temporally coordinated respiratory output from cervical and lumbar motor pools, as well as internal intercostal respiratory muscles. (**A**) Cartoon of sagittal view through caudal hindbrain indicating the locations of distinct developmentally defined glutamatergic populations important for breathing. Colored filled circles indicate the relative location of rhombic lip (RL) *Atoh1* (orange), RTN *Phox2b* (purple), and *Dbx1* (blue) derived neurons. Note: RTN *Phox2b* neurons transiently express *Atoh1*, but we have left this out for clarity. *Atoh1* RL neurons do not express *Phox2b* and have not been shown to be chemosensitive. The green circle identifies the location of preBötzinger Complex (preBötC) *Dbx1* neurons hypothesized to generate the cervical (inspiratory) rhythm. The magenta rectangle indicates the location of Retrotrapezoid/Parafacial respiratory group (RTN/pFRG) region neurons hypothesized to generate independent lumbar (expiratory) rhythm. LRN–lateral reticular nucleus, NA–nucleus ambiguus, rVRG–rostral ventral respiratory group, VIIn (VII motor nucleus). Scale bar = 200 µm. (**B**) Schematic describing the hypothesized glutamatergic coupling between *Dbx1* preBötC neurons generating inspiration (green circle–right) and the three candidate glutamatergic RTN/pFRG populations generating expiration (magenta rectangle–left). Colored circles indicate genetic lineage as in (**A**). (**C**) Electrophysiological traces (upper–integrated, lower–raw) of spontaneous respiratory-related output from cervical (C4, green) and lumbar (L1, magenta) motor roots as well as EMG recording from the XIth internal intercostal muscle (IC). Cervical

*Figure 1. Continued*

roots innervate the diaphragm active during inspiration. Lumbar roots innervate abdominal respiratory muscles active during expiration in adults. Arrows indicate respiratory-related bursts where some respiratory motor pools lack activity. (**D**) In some fictive breaths, cervical (green), lumbar (magenta), and IC (black) motor outputs are active nearly simultaneously. (**E**) In other fictive breaths only the cervical root is active. (**F**–**I**) Single integrated C-L-IC fictive breaths showing breath-by-breath variations in temporal co-activation between respiratory motor outputs. Note that in some respiratory bursts (**G**–**I**) both lumbar (magenta) and IC (black) burst peaks occur before the cervical burst. Also IC activity can occur during both cervical and lumbar bursts (**H**–**I**). (**J**–**K**) Cartoons indicating how different patterns of respiratory motor output from cervical and lumbar motor pools could be produced by changes in the synaptic strengths of excitatory (dots) or inhibitory (arrowheads) synaptic connections between the putative preBötC *Dbx1* inspiratory oscillator, the unknown RTN/pFRG expiratory oscillator, and intervening inhibitory interneurons. Solid lines indicate strong connections, dotted lines indicate weaker connection. Scale bar = 10 s (**C**), 500 ms (**D**–**I**).

The third RTN/pFRG population consists of glutamatergic neurons derived from *Atoh1*-expressing progenitors within the rhombic lip (hereafter called RL neurons), some of which express *LIM Homeobox 9* (*Lhx9*) post-mitotically (*Figure 1A,B, 2E,F*). These neurons require *Atoh1* for their initial neural specification (*Wang et al., 2005*), express the glutamate transporter, *solute carrier family 17 (sodium-dependent inorganic phosphate cotransporter), member 6* (*SLC17A6*), also known as the vesicular glutamate transporter 2 (*VGlut2*), and do not project to the cerebellum (*Wang et al., 2005*; *Gray, 2013*). Importantly, these neurons do not express Phox2b and have not been shown to be chemosensitive. The germline elimination of *Atoh1* in mice leads to the complete loss of RL neurons and abnormal migration and functional elimination of RTN neurons (*Rose et al., 2009b*; *Huang et al., 2012*). This produces a completely penetrant perinatal lethality due to the inability to establish respiratory rhythm in vivo, although reduced preparations containing the preBötC can still produce rhythmic inspiratory output (*Rose et al., 2009b*). In contrast, the selective elimination of functional RTN neurons by the conditional genetic targeting of *Atoh1* in subsets of hindbrain neurons expressing *Phox2b* does not eliminate breathing in vivo but produces partial neonatal lethality and blunting of chemosensitivity (*Dubreuil et al., 2009*; *Ramanantsoa et al., 2011*; *Huang et al., 2012*). These observations suggest that both RL and RTN neurons play important roles in breathing.

We used multiple transgenic mouse lines to test the role of distinct brainstem populations in generating and coordinating respiratory behaviors in mice. We found that *Dbx1*-dependent, but not *Atoh1*-dependent, neurons are necessary for rhythmic motor output from respiratory motoneurons. RL neurons, however, are required for temporal delay and coupling between multiple respiratory oscillators. This suggests that genetically distinct populations independently control two fundamental aspects of motor behavior: rhythm generation and the relative timing between motor pools.

## Results

### E18.5 mouse hindbrain-spinal cord preparations produce coordinated respiratory motor outputs

Post-natal rodent hindbrain-spinal cord preparations generate robust respiratory-related rhythmic bursts (fictive breaths) from cervical (diaphragm) and lumbar (abdominal) motor roots, with abdominal output requiring the RTN/pFRG region (*Smith et al., 1990*; *Iizuka, 1999*, *2004*; *Janczewski and Feldman, 2006*; *Taccola et al., 2007*; *Abdala et al., 2009*). These outputs are hypothesized to represent in vitro correlates of inspiratory (cervical) and expiratory (lumbar) motor output (*Abdala et al., 2009*; *Feldman et al., 2013*). Because mouse mutants affecting *Atoh1*, *Phox2b*, or *Dbx1* neurons either do not breathe, or have respiratory depression in vivo, we first tested whether the entire hindbrain and spinal cord at E18.5 from timed pregnant wild-type mice produced coordinated cervical and lumbar output. Previous work has shown rhythmic cervical (inspiratory) output beginning around E15 in mice, but whether lumbar (expiratory) activity is also present is unknown (*Thoby-Brisson et al., 2005*).

We simultaneously recorded endogenous respiratory-related cervical and lumbar root output as well as electromyographic (EMG) output from the XIth internal intercostal muscle (IC). The IC muscle is innervated by motoneurons in the caudal thoracic spinal cord. Like the lumbar innervated abdomen, the IC is active during expiration in vivo and in more mature in vitro preparations (*Miller et al., 1985*,

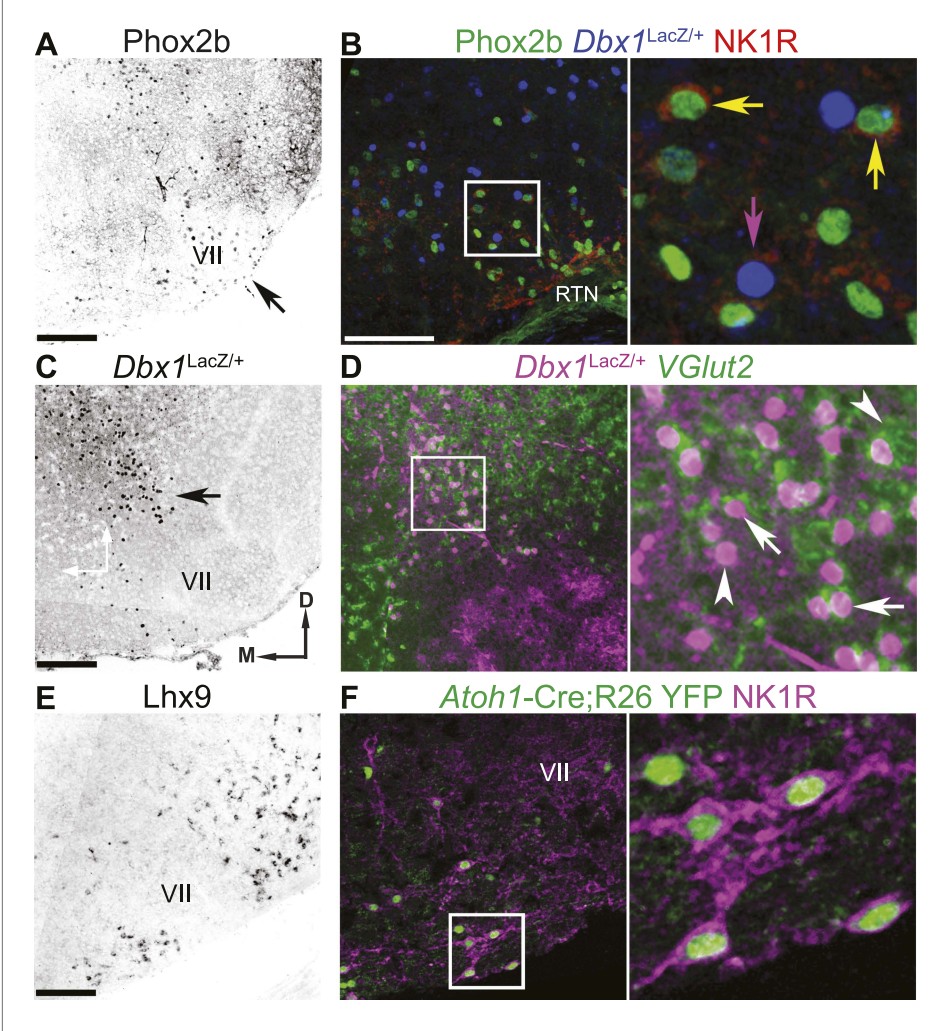

**Figure 2**. The RTN/pFRG contains three candidate expiratory rhythm generating populations. (**A** and **B**) *Phox2b*-expressing RTN neurons are located directly ventral to VII motor nucleus (VII) (**A**–arrow, **B**–green); many co-express NK1R (**B**, red). (**C** and **D**) *Dbx1*-dependent LacZ-labeled neurons are located ventral and dorsomedial to VII (**C**–arrow). RTN/pFRG *Dbx1*-derived LacZ-expressing neurons (**B**–blue, **D**–magenta) co-express NK1R (**B**–red) and *VGlut2* mRNA (**D**–green) in a P0 *Dbx1*LacZ/+ mouse (**B**–**D**). (**E** and **F**) *Atoh1*-dependent RL neurons are located medial and ventral to VII, and many express *Lhx9* mRNA (**E**) or NK1R (**F**–magenta) in P0 WT (**E**) or *Atoh1*-CreTg;R26 YFP (**F**–green) mice. Arrows in **B** and **D** indicate co-expression, arrowheads indicate lack of co-expression. Squares (**B**, **D**, **F**) are enlarged to right. Scale bars = 200 μm. D-dorsal, M-medial. See *Figure 2—figure supplement 1* for the effects of loss of *Atoh1* on RTN neurons as well as the loss of *Dbx*1 on RTN/pFRG *VGlut2* expressing neurons.

The following figure supplements are available for figure 2:

**Figure supplement 1**. Loss of *Atoh1* or *Dbx1* eliminates specific RTN/pFRG neuronal populations.

*1987*; *Iscoe, 1998*; *Legrand and Troyer, 1999*; *Iizuka, 2003*, *2004*). As in preparations from older animals, we found rhythmic motor output from both cervical and lumbar roots that corresponded to rhythmic respiratory muscle activity (*Figure 1C*). These outputs were silenced by transection above the first cervical root (not shown), and thus require the hindbrain. In late fetal (E18.5) preparations, however, both lumbar motor roots and intercostal EMG were often co-active with cervical output (*Figure 1C,D,F*). Importantly, however, cervical, lumbar, and IC activities were not identical arguing against a single source of respiratory drive for all respiratory motor pools. Cervical bursts were present without lumbar motor root or intercostal (IC) muscle activity, as is most common in adult rodents at rest (see arrows in *Figure 1C*, *Figure 1E*), whereas both lumbar and IC activity were often active prior to

the onset of cervical output, as in older preparations (*Figure 1G–I*). These data are consistent with the presence of functional and independent rhythmogenic sources of respiratory drive to cervical, thoracic, and lumbar motor pools in E18.5 mice. It further suggests these different patterns of temporal co-activation from different respiratory motor pools might be the consequence of variations in the coupling between independent respiratory oscillators (*Figure 1J–L*).

## *Dbx1*-dependent, but not RL or RTN, neurons are necessary for respiratory rhythm generation

While these data are consistent with the coupling of a functional RTN/pFRG with the preBötC to produce coordinated cervical–lumbar motor output, the genetic identity of the essential neurons for lumbar output was unknown (*Figures 1B and 3A*). We set out to directly test the role of RTN and RL neurons in the generation and coordination of respiratory motor activity by genetically eliminating specific hindbrain populations in transgenic mice. RTN neurons are functionally eliminated (RTN⁻) by targeted deletion of *Atoh1* in *Phox2b*-expressing neurons in *Phox2b*-Cre; *Atoh1*$^{LacZ/F}$ mice (RL⁺/RTN⁻) (*Figure 2—figure supplement 1A,B*; *Huang et al., 2012*). Because the loss of *Atoh1* does not completely ablate RTN neurons, we also examined *Phox2b*-Cre; *VGlut2*$^{F/F}$ mice in which *Phox2b* excitatory neurons are selectively disconnected from hindbrain networks (RL⁺/RTN$^{silent}$) (*Rossi et al., 2011*). In both of these lines, preBötC and other hindbrain *Dbx1*-dependent neurons remain unaffected (*Rose et al., 2009b*; *Gray et al., 2010*; *Huang et al., 2012*; *Feldman et al., 2013*).

The loss of RTN neurons leads to respiratory depression in vivo and in vitro, and RTN *Phox2b* neurons have been a leading candidate for an independent respiratory oscillator driving abdominal and, possibly, VII motor output (*Onimaru et al., 2008*; *Thoby-Brisson et al., 2009*; *Feldman et al., 2013*). WT hindbrain-spinal cord preparations produce highly reliable rhythmic cervical and lumbar motor outputs (*Figures 1C and 3B*). Surprisingly, both cervical and lumbar respiratory motor outputs were still present after either elimination or silencing of RTN neurons in RL⁺/RTN⁻ and RL⁺/RTN$^{silent}$ mice, although the cycle time between successive fictive breaths (the respiratory period) dramatically increased (p<0.01 in RL⁺/RTN⁻ mice, *Figure 3C,D,G*) (*Dubreuil et al., 2009*; *Ramanantsoa et al., 2011*; *Huang et al., 2012*). Substance P (SP) application in RL⁺/RTN⁻ mice recovered inspiratory periods to WT levels (p<0.0001, *Figure 3G*), likely due to direct effects upon preBötC neurons and other RTN/pFRG neurons (*Figure 2B,F*; *Gray et al., 1999*).

Given that lumbar motor output persisted after RTN loss, we explored whether the combined loss of both RTN and RL neurons in *Atoh1*$^{LacZ/LacZ}$ mice (RL⁻/RTN⁻) would affect respiratory behaviors (*Figure 3E*). We again found that rhythmic respiratory output from both cervical and lumbar rhythms persisted, but the combined loss of both RL and RTN recovered respiratory periods to near WT levels under both baseline conditions and with SP application (*Figure 3G*). Lastly, we tested the effect on lumbar output of the complete loss of *Dbx1*-dependent neurons in *Dbx1*$^{LacZ/LacZ}$ mice (*Dbx1*⁻, *Figure 3F*), where both RL and RTN populations are unaffected (*Bouvier et al., 2010*; *Gray et al., 2010*). *Dbx1*-derived neurons are essential for the formation of the preBötC and rhythmic cervical output, but their role in lumbar output was unknown. In contrast to mice with loss of RL and/or RTN neurons, *Dbx1*⁻ mice produced no rhythmic respiratory output from either cervical or lumbar motor roots, even when stimulated with SP or elevated potassium in the hindbrain perfusion solution (*Figure 3F*). This failure was not due to an inability to generate spinal output of any kind, because global disinhibition of the hindbrain by application of bicuculline and strychnine produced robust but non-respiratory spinal outputs (not shown). These results are consistent with an important role for RTN neurons in modulating breathing, but undermine current hypotheses by indicating that RTN and RL neurons are neither necessary nor sufficient for lumbar respiratory motor output (*Onimaru et al., 2008*; *Thoby-Brisson et al., 2009*; *Mellen and Thoby-Brisson, 2012*). *Dbx1*-derived neurons, however, are essential for the expression of both cervical and lumbar respiratory behaviors (*Bouvier et al., 2010*; *Gray et al., 2010*).

## Independent oscillators persist in *Atoh1* mutant mice

The maintenance of lumbar output in RL⁻/RTN⁻ mice raises the question of whether the networks underlying respiratory outputs to different motor pools in WT mice are generated by multiple oscillators (*Mellen et al., 2003*; *Mellen and Thoby-Brisson, 2012*; *Hägglund et al., 2013*; *Moore et al., 2013*) or by interactions within a single oscillatory population (*Smith et al., 2007*). In WT mice, most fictive breaths showed cervical and lumbar motor co-activation. Often, however, individual cervical respiratory bursts occurred without a corresponding lumbar burst, and vice versa (*Figure 1E*, arrows

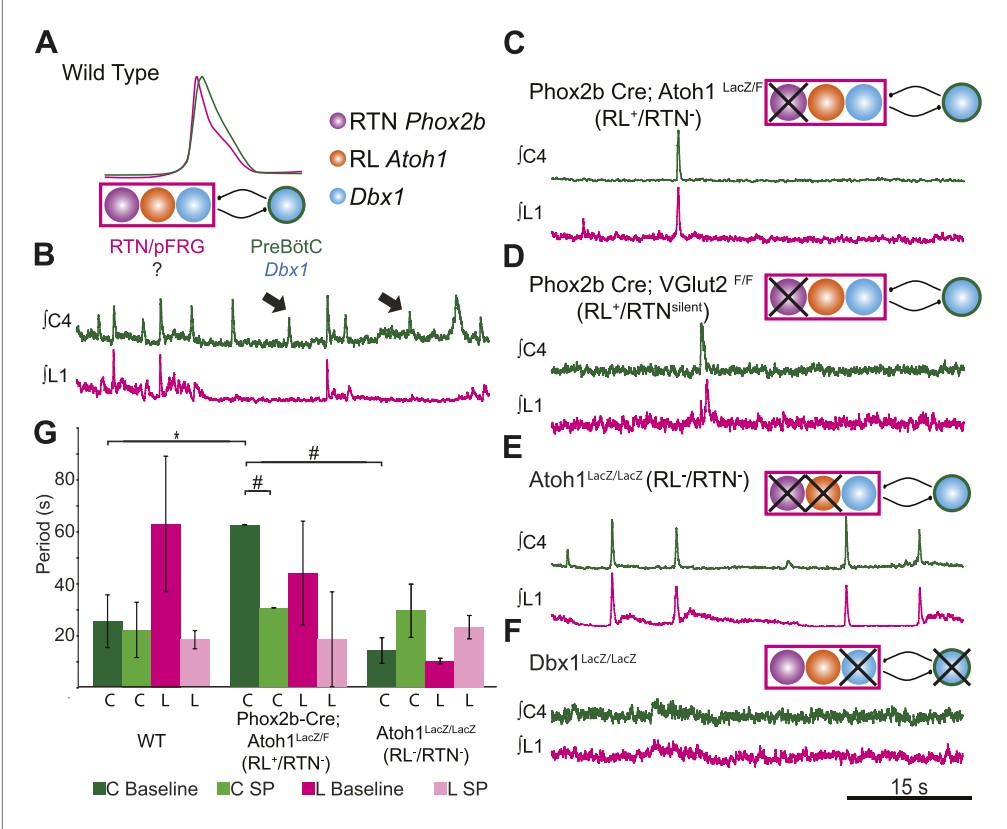

**Figure 3**. RL and RTN neurons are neither necessary nor sufficient for lumbar respiratory output. (**A**) Schematic cartoon describing populations targeted for genetic elimination to determine necessity for generating cervical (green) and/or lumbar (magenta) respiratory output. Colored filled circles indicate the developmental origin of RL *Atoh1* (orange), RTN *Phox2b* (purple), or *Dbx1* (blue) derived neurons with schematic as in *Figure 1B*. (**B**) 50 s integrated recordings of spontaneous respiratory output from cervical (C4, green) and lumbar (L1, magenta) motor roots in an E18.5 WT isolated hindbrain-spinal cord preparation showing normal lumbar-cervical coordinated respiratory output. Upward deflections indicate respiratory-related events. Arrows indicate fictive breaths where cervical output is not matched by lumbar output. Targeted ablation (**C**, *Phox2b*-Cre;*Atoh1*LacZ/F [RL+/RTN−]) or silencing (**D**, *Phox2b*-Cre;*VGlut2*F/F [RL+/RTNsilent]) of RTN *Phox2b* neurons does not eliminate lumbar respiratory output. Schematics (**C**–**D**, upper right) indicate targeted loss of *Phox2b* RTN neurons (purple). Note the marked increase in respiratory period (i.e., slowing of frequency). (**E**) Loss of both RTN *Phox2b* and RL *Atoh1* neurons in *Atoh1*LacZ/LacZ (RL−/RTN−) mice similarly does not eliminate lumbar respiratory output. Schematic (upper right) indicates targeted loss of both *Phox2b* RTN (purple) and RL *Atoh1* neurons (orange). (**F**) Loss of *Dbx1*-dependent neurons in *Dbx1*LacZ/LacZ mice eliminates both lumbar and cervical respiratory outputs. Schematic (upper right) indicated targeted loss of *Dbx1* neurons (blue). Scale bar = 15 s. (**G**) Loss of RTN Phox2b neurons slows the respiratory rhythm, thus increasing the cervical period. Bar graphs show least-squares mean (LSMEAN) of respiratory periods in seconds (±SEM) during baseline (cervical–[C], dark green, lumbar–[L], magenta) or in the presence of 1 µm SP (cervical–[C], light green, lumbar–(L), pink) in wild-type (WT) (n = 6), *Phox2b*-Cre;*Atoh1*LacZ/F (RL+/RTN−) (n = 4), and *Atoh1*LacZ/LacZ (RL−/RTN−) (n = 5) mice. Top brackets indicate statistical test groups (*p<0.05, #p<0.001; mixed random effects ANOVA).

in *Figures 1C and 3B*). Respiratory bursts in which one or more motor outputs are absent (termed deletions) occur both in vivo and in vitro (*Iizuka, 2010*; *Pagliardini et al., 2011*). Analysis of deletions is used in spinal motor circuits as a test of independent oscillators controlling antagonist flexor and extensor muscles (*McCrea and Rybak, 2008*; *Stein, 2008*; *Grillner, 2011*). In general, a single rhythmogenic source drives multiple motor outputs nearly identically. We reasoned that decreases in the number of oscillators driving motor output in mutant mice should be revealed by large decreases in, if not elimination of, respiratory deletions. This should be most true of lumbar motor outputs hypothesized to be generated by RTN/pFRG neural populations. We therefore determined the

percentage of respiratory bursts in which respiratory motor pools were co-active in WT and *Atoh1* mutant mice.

We recorded simultaneously from three respiratory motor roots, lumbar, cervical and cranial facial (VII). VII motoneurons are directly adjacent to the location of the hypothesized secondary oscillator and can show rhythmic motor output in the absence of the preBötC (*Onimaru et al., 2006*). WT cervical-lumbar-VII bursts also showed highly associated output (*Figure 4A*). The outputs from these three roots were not identical, however, as one might assume if they were driven by a single rhythmic source. Cervical bursts, for example, were co-active with both lumbar and VII (C+L+VII) only 29.2 ± 9% of the time (n = 4, *Figure 4C,D,G*, *Figure 4—figure supplement 1B*). Lumbar and VII bursts were co-active with both other roots (C+L+VII) 34.9 ± 11.7% or 46.7 ± 11.3% of the time respectively (*Figure 4C,D,G*, *Figure 4—figure supplement 1A–D*; *Table 1*). All three motor pools could be active either alone or with one or the other motor pool, suggesting a high degree of independence (*Figure 4C,D,G*, *Figure 4—figure supplement 1A–D*). Cervical, lumbar, or VII roots were active independently, that is only a single motor pool was active, 23 ± 5.8%, 14 ± 6.3%, or 18.6 ± 10.4% of the time, respectively (*Figure 4G*; *Table 1*). This partial independence of motor output was also seen in cervical-lumbar-IC triple recordings, where 88.0 ± 5.1% of IC muscle contractions were co-active with both cervical and lumbar output (*Table 2*).

Because the WT results indicated that multiple respiratory motor pools received similar but not identical respiratory drives which varied on a breath-by-breath basis (*Carroll and Ramirez, 2013*), we went on to test the effect of eliminating RTN and RL populations on respiratory deletions. Similar to WT, in RL−/RTN− mice 30.9 ± 7.9% of cervical bursts showed lumbar and VII co-activation (C+L+VII, i.e., no deletion), as compared to 33.2 ± 10.6% of lumbar and 74 ± 2.5% of VII bursts showing triple co-activation (*Figure 4B,E–H*, *Figure 4—figure supplement 1E–L*; *Table 3*). In some cases, we observed bouts of multiple non-overlapping respiratory bursts over short timescales (*Figure 4H*). We also observed independent, higher frequency lumbar-only bursts as well as the maintenance of cervical and lumbar deletions in RL+/RTN− mice under baseline conditions (*Figure 4—figure supplement 2A–C*). Overall, there was no consistent decrease in deletions amongst all three motor roots between control and RL−/RTN− mice. Lumbar only bursts showed a 24.3% increase while cervical only bursts decreased by 11.7% and independent VII bursts decreased by 51.1% although independent VII outputs were still present (*Table 4*). Similar variations in two or three root co-activation were also seen (*Table 4*).

The maintenance of independent output patterns from all three motor roots in RL−/RTN− and the presence of independent motor output with different frequencies are evidence for the maintenance of separate extant oscillators. Our present results also match invertebrate preparations where individual oscillators increase their frequency after uncoupling from a network (*Mulloney, 1997*). These data suggest that, in WT animals, each respiratory motor pool receives drive from independent but interconnected oscillators and that these oscillators persist in RL−/RTN− mice. These data further suggest that the loss of both RTN and RL neurons accentuates free-running oscillator activity induced by selective neuromodulation (*Doi and Ramirez, 2008*, *2010*).

## *Atoh1*-dependent neurons are necessary for respiratory-related biphasic cervical bursts

The primary neurotransmitter of both RL and RTN neurons is glutamate (*Stornetta et al., 2006*; *Rose et al., 2009a*). Consistent with their stimulatory role, the selective loss of RTN neurons leads to a significant prolongation of the inspiratory period. After the loss of both RL and RTN populations, however, the respiratory period recovered almost to WT levels (*Figure 3C–E,G*). This raised the possibility that RL neurons mediate a net inhibitory effect on respiratory networks, possibly via activation of inhibitory neurons. Inhibition is essential for normal respiratory output to control the timing and pattern of the different respiratory muscles (*Marder and Bucher, 2001*; *Ramirez and Viemari, 2005*; *Feldman et al., 2013*; *Janczewski et al., 2013*). The possibility of a net inhibitory role of RL neurons led us to examine the temporal structure of respiratory motor output in WT and RL−/RTN− mice.

Respiratory networks produce several motor behaviors besides the basic cervical (eupneic) respiratory pattern. The best characterized of these are sighs (*Ramirez and Viemari, 2005*), which are generated within the medulla, persist in reduced preparations, are first seen during late embryonic stages, and are increased after SP application (*Lieske et al., 2000*; *Gray et al., 2001*; *Ballanyi and Ruangkittisakul, 2009*; *Chapuis et al., 2014*). Sighs consist of a biphasic inspiratory (cervical) burst with an initial peak of typical amplitude that, before decaying, gives rise to a larger amplitude sigh

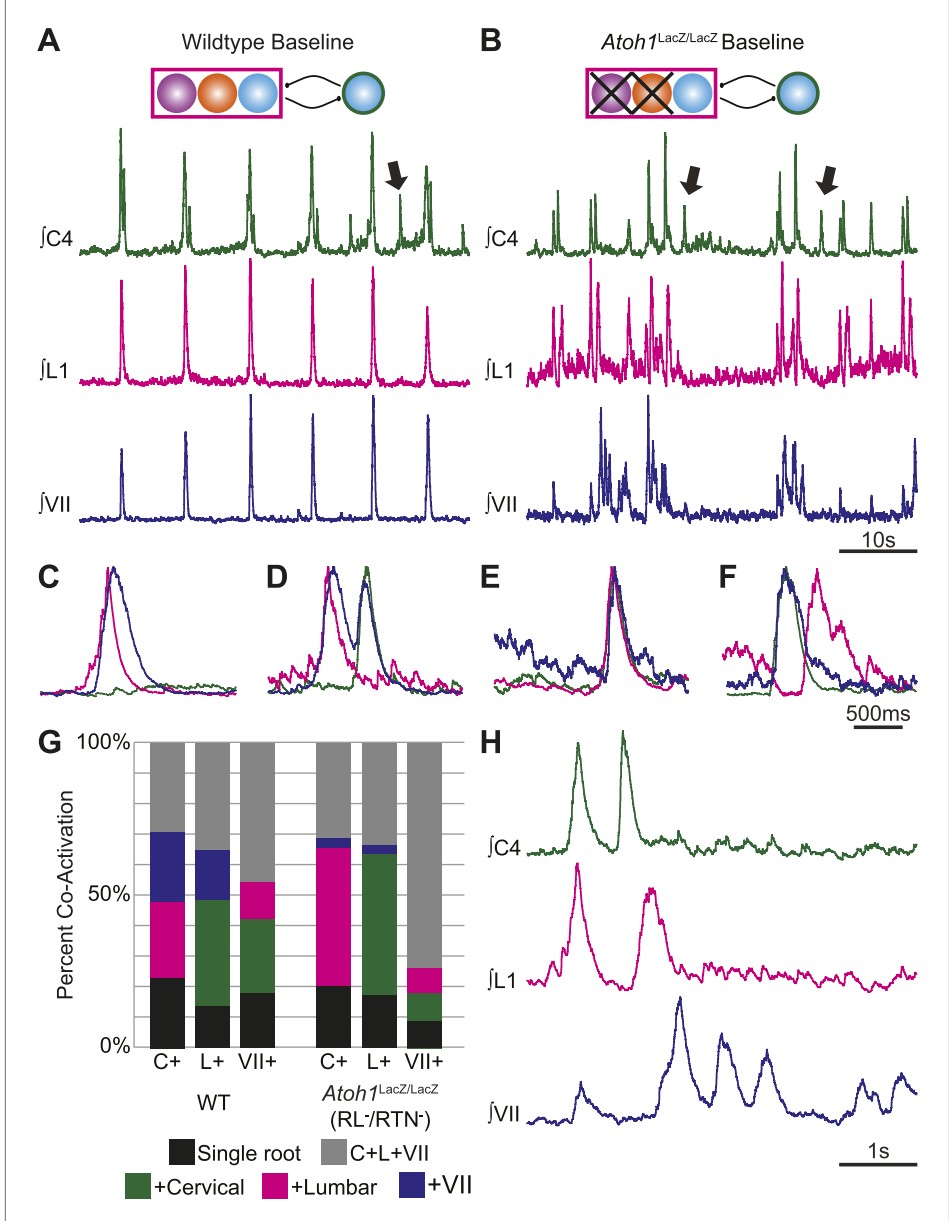

**Figure 4**. RL and RTN neurons are not essential for the expression or independence of cervical, lumbar, or VII respiratory motor outputs. (**A**) E18.5 WT mice show respiratory co-activation of cervical (green), lumbar (magenta), and/or VII (dark blue) motor roots. Schematic (top) indicates maintenance of all RTN/pFRG and preBötC glutamatergic lineages. (**B**) Targeted loss of RL and RTN neurons in *Atoh1*^LacZ/LacZ (RL⁻/RTN⁻) mice does not eliminate respiratory co-activation of cervical, lumbar, and/or VII respiratory outputs. Schematic (top) indicates targeted loss of *Phox2b* RTN (purple) and *Atoh1* RL (orange) neurons. Arrows (**A** and **B**) indicate respiratory cervical outputs lacking lumbar or VII output. During single respiratory bursts, each motor pool can be co-active nearly simultaneously, with a temporal delay, or can be silent in relation to each other motor pool in WT (**C** and **D**) or in the absence of both RTN and RL neurons (**E** and **F**). (**G**) Stacked histograms showing the percentage of fictive breaths from each individual motor pool (cervical–C+, lumbar–L+, VII–VII+) where that motor pool fires either alone (bottom–black), fires with one of the two other motor roots (middle two bars, +cervical–green, +lumbar–magenta, +VII–dark blue), or where all three roots are co-active (top, C+L+VII–gray) from WT (left) or in *Atoh1*^LacZ/LacZ (right) mice. Note the loss of RL and RTN neurons only affects the relative percentage of coupling. Scale bars = 10 s (**A** and **B**), 500 ms (**C**–**F**), 1 s (**H**). See *Figure 4—figure supplements 1A–L* for additional examples of the independence of respiratory motor pool co-activation in WT and *Atoh1*^LacZ/LacZ mice. See *Figure 4—figure supplements 2A–C* for examples of independent

*Figure 4. Continued*

lumbar activation and quantification of cervical and lumbar deletions in *Phox2b*-Cre;*Atoh1*$^{LacZ/F}$ (RL$^+$/RTN$^-$) mice.

The following figure supplements are available for figure 4:

**Figure supplement 1**. Maintenance and independence of multiple respiratory motor outputs in RL$^-$/RTN$^-$ mice.

**Figure supplement 2**. Independent lumbar oscillator is transiently released in RL$^+$/RTN$^-$ mice.

---

**Table 1.** Baseline percentage of simultaneous co-activation for zero, one, or two other motor pools between cervical, lumbar, and VII respiratory outputs in E18.5 WT mice

| | Percentage of Co-activation | | | |
| | +Cervical | +Lumbar | +VII | C+L+VII |
|---|---|---|---|---|
| Cervical | 23 ± 5.8 | 25.1 ± 11.6 | 22.8 ± 9.7 | 29.2 ± 9 |
| Lumbar | 34.7 ± 17.5 | 14 ± 6.3 | 16.4 ± 8.4 | 34.9 ± 11.7 |
| VII | 24.9 ± 9 | 12.3 ± 4.5 | 18.6 ± 10.4 | 46.7 ± 11.3 |

---

**Table 2.** Baseline percentage of simultaneous co-activation with zero, one, or two other motor pools between cervical, lumbar, and intercostal XI (ICX1) respiratory outputs in E18.5 WT mice

| | Percentage of co-activation | | | |
| | +Cervical | +Lumbar | +IC | C+L+IC |
|---|---|---|---|---|
| Cervical | 27.4 ± 9.6 | 7.1 ± 3.7 | 7.1 ± 2.8 | 58.5 ± 11.6 |
| Lumbar | 12.5 ± 7.6 | 3.2 ± 1.5* | 0.0 | 84.3 ± 7.2 |
| IC | 10.4 ± 4.5 | 0.0 | 1.6 ± 0.8† | 88.0 ± 5.1 |

*p<0.026.
†p<0.018.
One-way ANOVA vs cervical.

---

**Table 3.** Baseline percentage of simultaneous co-activation for zero, one, or two other motor pools between cervical, lumbar, and VII respiratory outputs in E18.5 *Atoh1*$^{LacZ/LacZ}$ (RL$^-$/RTN$^-$) mice

| | Percentage of co-activation | | | |
| | +Cervical | +Lumbar | +VII | C+L+VII |
|---|---|---|---|---|
| Cervical | 20.3 ± 1.8 | 45.6 ± 8.5 | 3.3 ± 0.8* | 30.9 ± 7.9 |
| Lumbar | 46.3 ± 7.8 | 17.4 ± 7.3 | 3.2 ± 1† | 33.2 ± 10.6 |
| VII | 8.9 ± 3.1 | 8.1 ± 3.3 | 9.1 ± 5.2 | 74 ± 2.5‡ |

*p<0.04, vs cervical.
†p<0.031 vs lumbar.
‡p<0.023 vs lumbar.
One-way ANOVA.

---

burst. This biphasic burst pattern is present under baseline conditions, both in vivo and in vitro. The sigh component of these biphasic motor outputs, however, can be decoupled from normal 'eupneic' bursts either pharmacologically or by blockade of inhibition (*Lieske et al., 2000*; *Ramirez and Viemari, 2005*; *Tryba et al., 2008*; *Chapuis et al., 2014*). Our WT E18.5 preparations showed biphasic cervical bursts similar to sighs (*Figure 5A,B,D,E*, *Figure 5—figure supplement 1A,B,F,G,I*). Under baseline

**Table 4.** Percentage change between WT and E18.5 *Atoh1*[LacZ/LacZ] (RL−/RTN−) mice of baseline simultaneous co-activation for zero, one, or two other motor pools between cervical, lumbar, and VII respiratory outputs

| | Percentage change of co-activation between WT and E18.5 *Atoh1*[LacZ/LacZ] (RL−/RTN−) mice. | | | |
| --- | --- | --- | --- | --- |
| | +Cervical | +Lumbar | +VII | C+L+VII |
| Cervical | −11.7 | 81.7 | −85.5 | 5.8 |
| Lumbar | 33.4 | 24.3 | −80.5 | −4.9 |
| VII | −64.3 | −34.1 | −51.1 | 164.7 |

conditions, 6 of 11 preparations produced biphasic cervical bursts (54.5%). In the presence of 1 μM SP, biphasic cervical bursts increased and were present in 3 of 4 preparations (75%). Alternately, biphasic bursts decreased in the presence of $10^{-12}$ M SST and were present in only 3 of 7 (42.9%) preparations. This is lower than what was recently found in E18.5 mouse in vitro slice preparations but not unreasonable given the elevated potassium used to stimulate respiratory-related output in slices (*Chapuis et al., 2014*). Under baseline conditions in preparations showing biphasic cervical bursts, these augmented bursts occurred in 8.87 ± 3.16% of total cervical bursts. In addition, biphasic cervical bursts also showed a consistent biphasic lumbar pattern in which activity occurred primarily during the initial cervical peak (*Figure 5A,D*, *Figure 5—figure supplement 1A,B,F,G,I*). SP increased the frequency of biphasic cervical bursts, which were often accompanied by an active inhibition of lumbar output during the second cervical peak (*Figure 5B*). This complex respiratory pattern was also found in cervical-lumbar-intercostal and cervical-lumbar-VII recordings (*Figure 5D,E*). In contrast to lumbar output, intercostal EMG activity was most active during the augmented burst, whereas VII output could be active during either burst (*van Lunteren et al., 1988*; *Lieske et al., 2000*; *Ramirez and Viemari, 2005*; *Figure 5D–E*). RL+/RTN− mice showed occasional biphasic cervical bursts, although lumbar activity in these events occurred during the larger peak (*Figure 5C*). Note that some cervical bursts that lack a biphasic component can show either smaller or larger amplitudes than putative sigh-bursts (*Figure 5—figure supplement 1C–E,H*).

In contrast, RL−/RTN− mice did not generate fictive breaths with biphasic cervical output (*Figure 5F,G*). They did show occasional double bursts, with two cervical outputs of similar amplitude separated by a short interval (*Figure 5H,I*). In these cases, lumbar activity was limited to the second cervical burst. In many cases, cervical-only bursts accompanied by active inhibition of baseline lumbar motor root activity were still present in both RL−/RTN− and RL+/RTN− mice (*Figure 4—figure supplement 1L*). This suggests that the absence of lumbar bursts can represent an active process similar to observations in vivo (*Pagliardini et al., 2011*) and is consistent with the maintenance of the neurons necessary for producing biphasic cervical outputs (not shown).

## Rhombic lip neurons are necessary for respiratory motor pool temporal delay

In adult mammals, abdominal muscles are usually silent during quiet breathing. Active abdominal activity during respiration is strongly state-dependent and appears under conditions of hypercapnia, hypoxia, anesthesia, and even sleep. When abdominal respiratory output is present, it usually precedes inspiratory activity, although in some cases it can also be present after inspiration (*Iizuka, 1999*; *Iizuka and Fregosi, 2007*; *Iizuka, 2009*; *de Almeida et al., 2010*; *Pagliardini et al., 2011*). In our E18.5 WT in vitro preparations, we noticed that lumbar activity either was co-active with, or preceded cervical activity (*Figure 6A–C*). Similar co-activation patterns were previously reported but not quantified in older in vitro rat preparations (*Taccola et al., 2007*). This suggested the temporal pattern of activation between motor pools might be controlled independently from the generation of each motor burst by unknown neurons. For each fictive breath, we calculated the time interval (τ) between the peaks of the integrated lumbar and cervical bursts (*Figure 6A*).

In E18.5 mice, lumbar bursts preceded cervical bursts with two distinct peaks (−220 or −60 ms, *Figure 6B,C*; evidence for both peaks is shown in the delay histogram (gray line in *Figure 6G*)). This is similar to intervals recorded in post-natal rat in vitro preparations (*Taccola et al., 2007*) and compares to delays of from less than 300 to over 500 ms between abdominal and diaphragm muscle activation in anesthetized adult rats breathing elevated $CO_2$ or decreased $O_2$ at 37° (*Iizuka and Fregosi, 2007*). Separately analyzing cycles based on long or short intervals, we found that the peak of lumbar output in our preparations preceded peak cervical output by ≥150 ms in 25.9 ± 6.6% of respiratory bursts (arithmetic mean) (n = 21, *Figure 6B,C,G,H*). In many cases, peak lumbar activity overlapped low-level

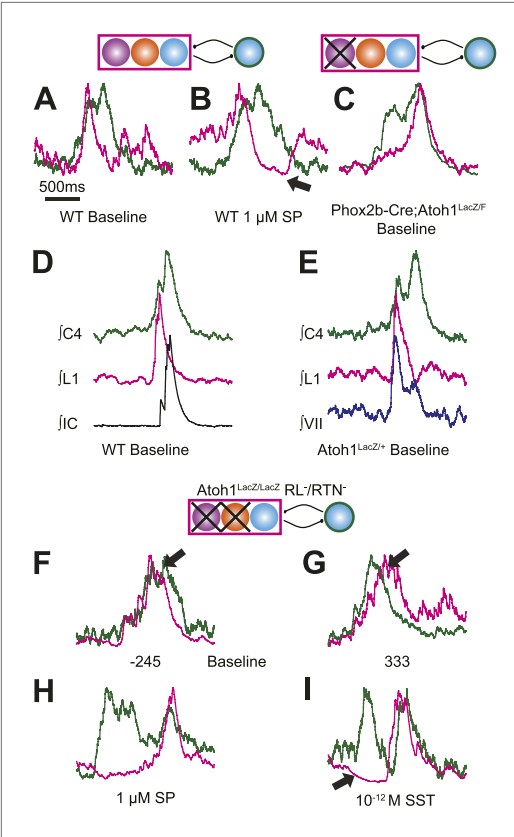

**Figure 5**. *Atoh1* neurons are necessary for biphasic cervical sigh-like fictive breaths. E18.5 WT mice produce biphasic respiratory-related cervical bursts (C4, green) with lumbar (L1, magenta) activation only during the initial normal amplitude cervical burst under baseline (**A**) or in the presence of 1 μM SP (**B**). Arrow in **B** indicates likely active inhibition of lumbar root during the larger amplitude cervical burst peak. (**C**) Biphasic cervical bursts persist after loss of RTN neurons in *Phox2b*-Cre;*Atoh1*LacZ/F (RL+/RTN−) mice with the lumbar burst occurring during the larger amplitude burst. Schematics indicate maintenance (top left) or targeted loss (right) of *Phox2b* RTN (purple) populations in WT or *Phox2b*-Cre;*Atoh1*LacZ/F (RL+/RTN−) mice. (**D–E**) In contrast to lumbar respiratory output (magenta), XIth internal intercostal (**D**, IC, black), or VIIn (**E**, VII, dark blue) can occur during both the initial and larger amplitude cervical burst (green) in E18.5 WT (**D**) or control *Atoh1*LacZ/+ heterozygote (**E**). Note the co-activation of cervical and VII roots during lumbar inhibition (**E**). (**F–I**) *Atoh1*LacZ/LacZ (RL−/RTN−) mice do not exhibit biphasic respiratory cervical bursts. Schematic indicates targeted loss of *Phox2b* RTN (purple) and *Atoh1* RL (orange) neurons. (**F–G**) Single integrated traces showing temporal separation between cervical (green) and lumbar (magenta) peaks due to increased noise (arrows). Numbers under traces indicate time of the lumbar peak in relation to the cervical peak (in ms). In some fictive breaths, *Atoh1*LacZ/LacZ mice show respiratory

*Figure 5. Continued on next page*

'pre-inspiratory' cervical activity (*Figure 6C*). Peptidergic modulation of the hindbrain with SP (1 μM), which stimulates preBötC and both RL and RTN populations, or SST ($10^{-12}$M), which inhibits preBötC but not RTN or rostral RL populations (*Gray et al., 2010*; *Gray, 2013*), did not statistically increase the percentage of respiratory bursts with temporal delay (36.9 ± 8.7% for SP modulation, p<0.6 n = 15; 39.6 ± 10.2% for SST modulation, p<0.6 [One-way ANOVA], n = 7, *Figure 6H*). These data indicate these reduced preparations can produce the adult-like respiratory pattern of lumbar activity preceding cervical activity (*Iizuka, 2011*).

In RL−/RTN− mice (n = 19), in contrast, the percentage of fictive breaths where the lumbar peak preceded the cervical peak by ≥150 ms was significantly decreased (4.1 ± 2.2%, p<0.004, Independent samples *t* test; *Figure 6D,G,I*). The interval between peaks in this 4.1% of respiratory bursts can be attributed to the inherent noise and variability in the output signals and are unlikely to reflect actual temporal disparity (*Figure 5F,G*). The temporal distribution for RL−/RTN− mice showed a single peak temporal difference that was significantly less than in WT (median peak lag in WT = −39 ms and RL−/RTN− mice = −3 ms, p<0.0016; *Figure 6G,I*). This loss of normal temporal interval was not a simple consequence of low excitability in the network, because neither excitation with SP nor preBötC inhibition with SST (*Onimaru et al., 2006*; *Gray et al., 2010*) generated any respiratory cycles with clear lumbar-cervical delay (*Figure 5F,G*, *Figure 6I*, *Figure 6—figure supplement 1A,B*). Unlike RL−/RTN− mice, however, RL+/RTN− and RL+/RTNsilent mice maintained the interval between the cervical and lumbar peaks in some fictive breaths (*Figure 6E,F*). The overall temporal distributions in RL+/RTN− affected preparations were not statistically different from those of either WT or RL−/RTN− mice because their rhythms were extremely slow and they had far fewer respiratory bursts to analyze. These data support previous findings that RTN neurons provide a baseline excitatory drive to maintain respiratory output (*Takakura et al., 2008*) but suggest that RL neurons have a different function, possibly related to relative timing between motor pools.

The change of the delay interval distribution between lumbar and cervical peaks in RL−/RTN− mice is consistent with a change in connectivity between independent oscillators. The appearance of uncoupled fictive breaths, especially evident in cervical-lumbar-VII recordings, led us to wonder whether what might appear to be deletions

*Figure 5. Continued*

doublets with two distinct cervical outputs with likely lumbar inhibition (arrow in **I**) during the initial burst in the presence of 1 µM SP (**H**) or 10⁻¹² M SST (**I**). Scale bar = 1 s. *Figure 5—figure supplement 1* shows the variability in amplitude and pattern of biphasic cervical bursts other respiratory bursts during baseline rhythmic activity as well as average and overlapping standard and biphasic bursts in a E18.5 WT mouse preparation.

The following figure supplements are available for figure 5:

**Figure supplement 1**. Normal and biphasic cervical respiratory bursts show different cervical and lumbar patterns and vary in amplitude during fictive breathing in an E185 mouse brainstem preparation.

at longer time scales (>1 s) were, in fact, only changes in the relative timing of respiratory output such that lumbar or VII activity occurs at a different phase of the respiratory cycle (*Figure 7A*). In vivo, the peak of activity for many respiratory neurons is delayed to late in the expiratory period (*Richter, 1982*). This type of phase switching is seen when moving from the simultaneous activation of legs during hopping to alternating activation during walking. To address this, we determined the phase of the peaks of lumbar and VII motor output during the cervical respiratory period (defined as Φ [0–360°] for each individual respiratory burst, *Figure 7A*) (*Strohl, 1985*; *Hwang et al., 1988*; *Plowman et al., 1990*; *Huangfu et al., 1993*; *Iscoe, 1998*). Consistent with our interval analysis in WT mice, we found a single peak for both lumbar and VII root activity

just prior to the peak of cervical output (*Figure 7B*). In RL⁻/RTN⁻ mice, the peak of both lumbar and VII activity also occurred just prior to cervical output (*Figure 7C*). Interestingly, VII output showed a broadening of the phase of activity relative to cervical output (*Figure 7C,D*). In WT mice we found no secondary peaks that would indicate consistent phase differences. In RL⁻/RTN⁻ mice, however, the lumbar trace showed a small second peak around 50° (arrows in *Figure 7C,E*). This peak represented a respiratory pattern unique to RL⁻/RTN⁻ mice that was characterized by an isolated cervical burst followed, after a pause, by lumbar and/or VII output (*Figure 7F*). Together, however, these data suggest an absence of a longer timescale reorganization of respiratory activity. They also show that both lumbar and VII outputs are still largely coupled to cervical output, with changes in the details of their relative timing.

## Proposed respiratory network

It is now possible to construct a model for the respiratory network that would explain the role of RL neurons (*Figure 8A*). We propose that RL neurons inhibit the preBötC via activation of inhibitory interneurons but excite the as-yet unidentified abdominal oscillator. The strength of RL-induced inhibition of preBötC neurons influences the temporal interval between cervical and lumbar output. Similarly, preBötC neurons activate inhibitory interneurons to attenuate lumbar output (*Pagliardini et al., 2011*). This type of asymmetrical inhibition has been suggested to underlie intersegmental coupling of independent oscillators in crayfish and lamprey rhythmogenic networks (*Mulloney, 1997*; *Hill et al., 2003*; *Mulloney and Hall, 2007*). This RL neuron-mediated net inhibition would also explain the remarkable slowing of respiratory-related rhythmic output seen with selective RTN ablation or silencing (*Figure 3C,D*).

To address the role of RL neurons, we recorded from fluorescently labeled RL, *Atoh1*-derived neurons in the ventrolateral medulla rostral to the preBötC but caudal to the RTN (*Figure 8B*). Consistent with our model, most RL neurons were tonically active (18 of 19, *Figure 8C,D*). One *Atoh1*-derived neuron was rhythmically inhibited during inspiration (*Figure 8E*). This inhibition reversed at potentials more negative than chloride reversal potential; it was eliminated by blockade of fast inhibitory amino acid receptors and consistent rhythmic synaptic input from inspiratory glycinergic neurons (*Figure 8F,G*; *Winter et al., 2009*; *Morgado-Valle et al., 2010*). Phasic inhibition suggests that at least some *Atoh1* neurons within the medulla are synchronized to respiratory output. These data suggest a mechanism by which the respiratory network can produce the variety of coordinated motor patterns seen in vivo (*Figure 8H–J*). Distinct but coupled oscillatory populations generate independent rhythms, but reconfiguration of the relative strengths of *Atoh1*- and *Dbx1*-dependent populations produces changes in the strength of both excitatory and inhibitory connections, leading to changes in respiratory pattern.

## Discussion

Breathing is a complex behavior that responds to the homeostatic needs of an organism and involves the coordinated activation of numerous respiratory muscles, which in mammals includes the diaphragm.

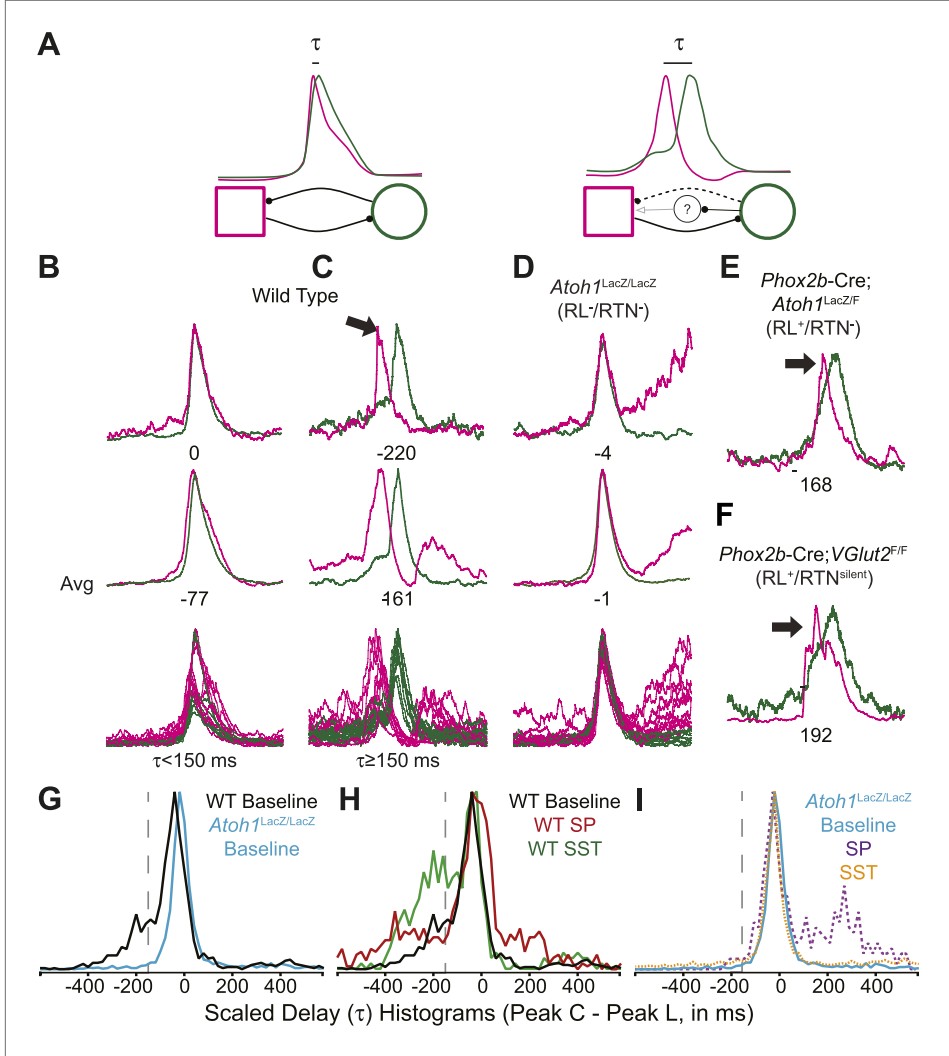

**Figure 6**. Rhombic lip, *Atoh1*-dependent neurons are necessary for normal cervical-lumbar temporal lag.
(**A**) Cartoons showing hypothesized network interactions (bottom) between putative independent lumbar (magenta square) and cervical (green circle) oscillators and unknown inhibitory interneurons, together generating two distinct temporal patterns of cervical (green) and lumbar (magenta) respiratory co-activation (top). These different networks produce motor outputs that differ in the relative time between their peaks (lumbar time-cervical time = τ, in ms).
(**B**–**D**) Integrated and overlapped cervical and lumbar traces from 10 consecutive (**B**, WT–τ <150 ms), sequential but not consecutive (**C**, WT–τ ≥150 ms), or consecutive (**D**, *Atoh1*^LacZ/LacZ^ [RL⁻/RTN⁻]) respiratory-related bursts aligned from the peak of cervical output (top–single burst, middle–average of 10 breaths, bottom–overlap of 10 bursts, numbers under traces = τ). Note the low level pre-inspiratory activity of the cervical trace during the lumbar burst as well as likely inhibition during the cervical burst in wild-type mice (**C**), but the strong simultaneous co-activation of lumbar and cervical outputs in *Atoh1*^LacZ/LacZ^ mice (**D**). Partial peak delays are still present in E18.5 *Phox2b*-Cre; *Atoh1*^LacZ/F^ (RL⁺/RTN⁻, **E**), and *Phox2b*-Cre;*VGlut2*^F/F^ mice (RL⁺/RTN^silent^, **F**) indicating *Phox2b* RTN neurons are not essential for temporal delay. (**G**–**H**) Histograms (12 ms bins) showing distributions of τ for WT (**G**–**H**) and *Atoh1*^LacZ/LacZ^ (**G** and **I**) mice to determine whether loss of RL and RTN neurons affects τ. Distributions are scaled to same height for better comparison. Gray dashed line indicates τ = −150 ms cutoff. Note the clear differences in the peak and spread between baseline WT and *Atoh1*^LacZ/LacZ^ distributions (**G**). WT but not *Atoh1*^LacZ/LacZ^ distributions show increased temporal delay after application of 1 µM SP or 10⁻¹²M SST (**H**–**I**). Histogram color: WT (baseline–black, SP–red, SST–green), Atoh1^LacZ/LacZ^ (baseline–cyan, SP–purple dashed, SST–orange dashed). See *Figure 6—figure supplement 1* for traces showing the absence of effect of SP or SST on temporal delay (τ) in Atoh1^LacZ/LacZ^ mice.

The following figure supplements are available for figure 6:

**Figure supplement 1**. Peptides do not induce cervical-lumbar temporal delay in *Atoh1*^LacZ/LacZ^ (RL⁻/RTN⁻) mice.

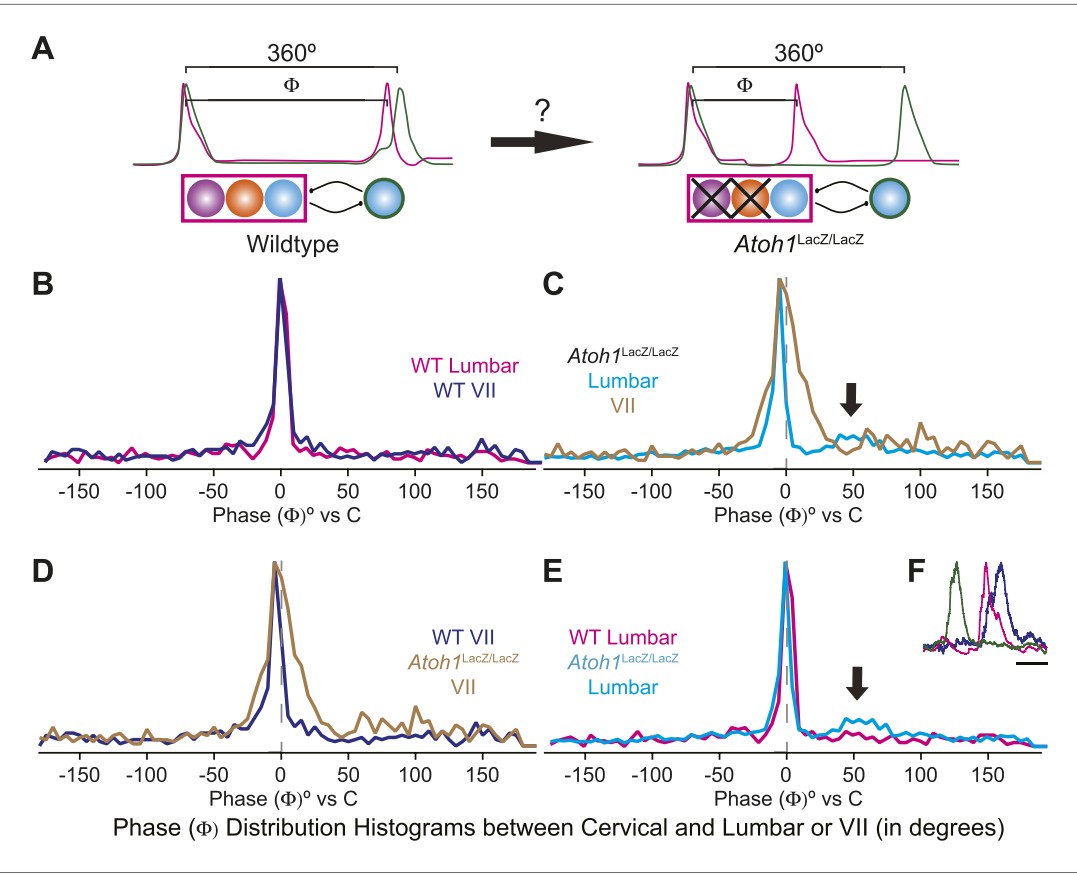

**Figure 7**. Loss of RL and RTN neurons does not change the phase relationship between putative respiratory oscillators. (**A**) Cartoons indicating how relative phase (Φ, in degrees) between cervical burst peaks and either lumbar or VII respiratory peaks for each fictive breath are calculated (left) and possible change in phase relationships after loss of RL and RTN neurons (right). Schematics (top right) indicate targeted loss of *Phox2b* RTN (purple) and *Atoh1* RL (orange) neurons. (**B**) Histogram (5° bins) showing the distributions of phase relationship (Φ) between cervical and either lumbar (magenta) or VII (dark blue) peaks indicating consistent temporal delays between outputs in WT mice. (**C**) Phase relationships (Φ) between cervical and either lumbar (cyan) or VII (tan) outputs indicate a maintenance in phase relationships between respiratory outputs in *Atoh1*LacZ/LacZ (RL⁻/RTN⁻) mice. (**D**) Loss of RL/RTN neurons broadens the phase relationship (Φ) of cervical-VII root activation compared to WT. (**E**). Loss of RL/RTN neurons does not effect phase (Φ) of most lumbar bursts but introduces a occasional *Atoh1*LacZ/LacZ specific burst pattern (arrow in **C** and **E**, **F**) with the cervical burst (green) followed by a pause then overlapping lumbar (magenta) and VII (dark blue) bursts. Scale bar = 500 ms.

The activities of other respiratory muscles are partially state-dependent (*Iscoe, 1998*; *Iizuka and Fregosi, 2007*; *de Almeida et al., 2010*; *Pagliardini et al., 2012*). Respiratory muscles, including the diaphragm, are also active during non-respiratory behaviors such as cough, emesis, valsalva, and hiccup (*Tomori and Widdicombe, 1969*; *Miller et al., 1987*; *Iscoe, 1998*; *Straus et al., 2003*). In this study, we begin to unravel how the respiratory network enables the temporal coordination of these different muscles.

In vitro preparations produce endogenous rhythmic output, which, while not exact replicates of breathing movements in vivo, have shed light on how respiratory, especially inspiratory, behaviors are generated (*Feldman et al., 2013*; *Funk and Greer, 2013*). Reduced in vitro slice and *en bloc* preparations, as well as in situ perfused brainstem preparations have been found to express a larger range of fictive respiratory behaviors including sighs, gasps, and active expiration (*Smith et al., 1990*; *Iizuka, 1999*; *Lieske et al., 2000*; *Shvarev et al., 2003*; *Iizuka, 2004*; *Taccola et al., 2007*; *Abdala et al., 2009*; *Funk and Greer, 2013*; *Chapuis et al., 2014*). These preparations have also allowed us to extend our analysis of respiratory networks to include transgenic mouse models that do not survive birth or early life (*Champagnat et al., 2011*).

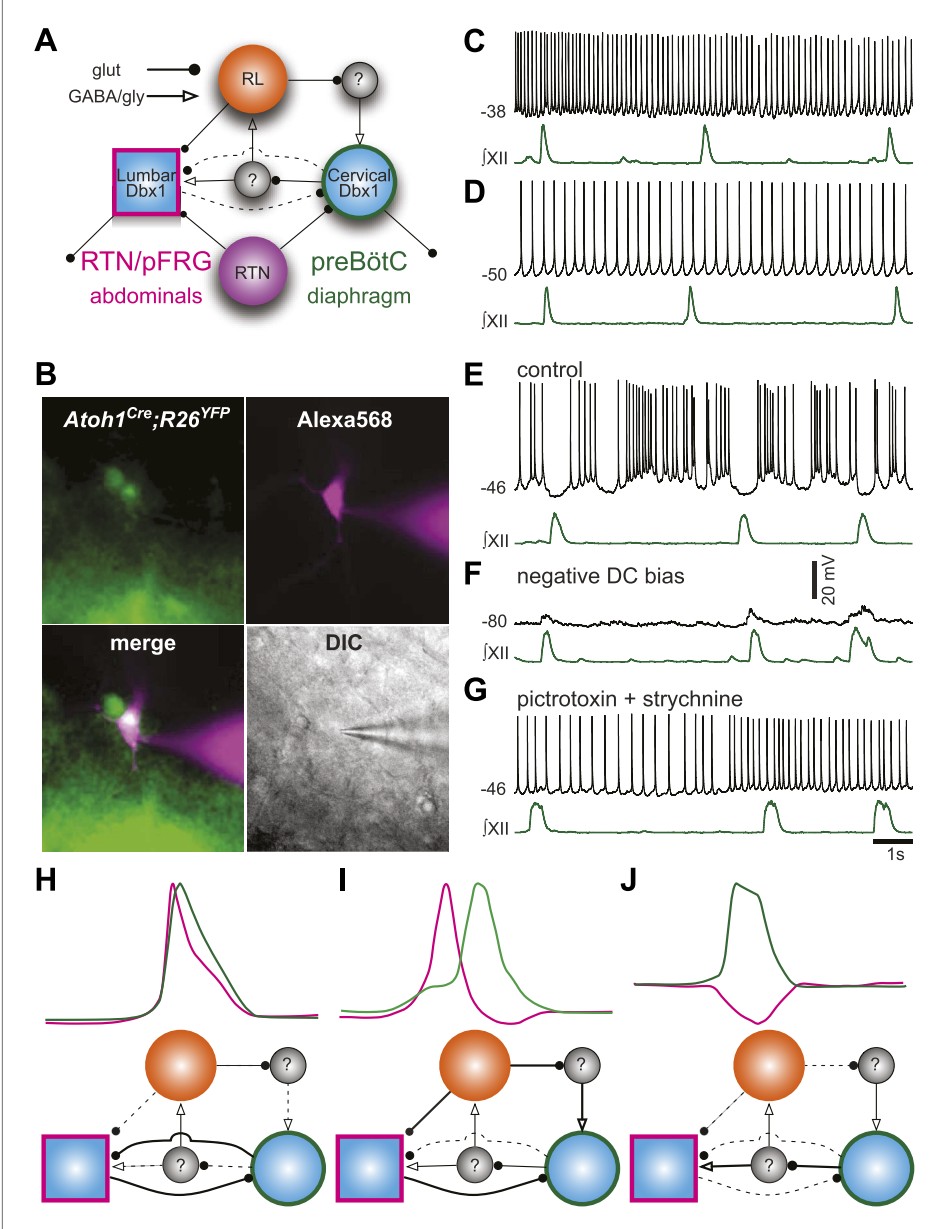

**Figure 8**. Proposed respiratory network underlying temporal relationship between inspiratory and expiratory respiratory oscillators. (**A**) Putative cervical (preBötC, green circle) and lumbar (RTN/pFRG, magenta square) oscillators contain *Dbx1*-derived neurons (blue filled circles) and drive expiratory (magenta, abdominals) and inspiratory (green, diaphragm) muscles. The preBötC is asymmetrically coupled to RTN/pFRG neurons via unknown inhibitory interneurons (gray filled circles). *Phox2b* RTN neurons (purple) provide excitatory drive to both oscillators. *Atoh1* RL neurons (orange) excite the RTN/pFRG oscillator. Lines with circles indicate glutamatergic connections. Lines with arrowheads indicate inhibitory GABA/glycine connections. Black lines indicate strong synaptic connections. Dashed lines indicate weak synaptic connections. (**B**) EYFP-labeled neurons (green) within the BötC area of an *Atoh1*-Cre;R26 EYFP slice preparation. The recorded neuron was filled with Alexa 568 (magenta) through whole-cell patch dialysis. The merged pseudo-color fluorescence and transmitted light IR-DIC images of the recorded neuron are also shown. (**C–F**) Current-clamp recordings of BötC *Atoh1* neurons (black, top). The corresponding integrated inspiratory activity is shown as ∫XII (bottom, green). The majority of *Atoh1* neurons show tonic non-respiratory activity (**C–D**). (**E**) Recording of an expiratory *Atoh1* neuron (from **B**) with rhythmic inhibition during inspiration. (**F**) Inhibitory inspiratory drive reverses with current bias to achieve a baseline membrane potential of −80 mV. Voltage scale bar = 20 mV. (**G**) Inhibitory inspiratory drive is eliminated with bath application of picrotoxin/strychnine. Scale bar = 1 s. (**H–J**) Cartoons showing hypothesized network reorganizations (bottom) underlying

*Figure 8. Continued on next page*

*Figure 8. Continued*

cervical (green) and lumbar (magenta) bursts (top). (**H**) Outputs without temporal lag are mediated by weak Atoh1 RL and inhibitory inputs and strong RTN/pFRG and preBötC excitatory coupling. (**I**) Fictive breaths with temporal delay between cervical and lumbar motor outputs have strong Atoh1 excitation of RTN/pFRG Dbx1 neurons and produce phasic inhibition of preBötC neurons. During the cervical burst, strong preBötC-mediated excitation of inhibitory interneurons leads to silencing of lumbar outputs. (**J**) Fictive breaths with lumbar inhibition during cervical bursts are due to weak Atoh1 RL neuron output and strong preBötC-mediated excitation of inhibitory interneurons leads to silencing of lumbar outputs. Lines define connectivity as in (**A**).

At late developmental stages, E18.5 WT mice delivered by C-section will breathe in vivo and produce robust inspiratory-related output in vitro (*Greer et al., 2006*). We found that at this age, in vitro preparations also produce robust rhythmic output from lumbar motor pools that innervate abdominal muscles active during expiration, thoracic motor pools that innervate intercostal muscles during expiration, and VII motor pools, all of which is consistent with previous work (*Huangfu et al., 1993*; *Iscoe, 1998*; *Onimaru et al., 2006*; *Iizuka, 2010*). In contrast to experiments in older animals, however, we found this lumbar output did not require altered pH or pharmacological stimulation. Moreover, we found the lumbar output overlapped cervical output for the majority of respiratory-related bursts, although the more mature 'pre-inspiratory' pattern was also present as in older rat in vitro preparations (*Taccola et al., 2007*).

Two developmentally distinct populations within the RTN/pFRG (*Phox2b-* and *Atoh1*-dependent RTN neurons) as well as *Atoh1*-dependent RL neurons have been hypothesized to generate the abdominal and/or VII rhythms (*Onimaru et al., 2008*; *Thoby-Brisson et al., 2009*; *Gray, 2013*). We find that both RL and RTN neurons are important for appropriately coordinated respiratory rhythm in cervical and lumbar motor pools. Respiratory periods are much longer in the absence of RTN glutamatergic neurons, consistent with previous work (*Dubreuil et al., 2009*; *Ramanantsoa et al., 2011*; *Huang et al., 2012*). The elimination of both RL and RTN neurons, however, recovered baseline respiratory period. Not only does this reveal a previously unknown net inhibitory role for RL neurons in the control of breathing, but it also demonstrates that neither of these populations is necessary or sufficient for expiratory motor output, since both cervical and lumbar respiratory-related outputs persist after their genetic elimination. In the absence of *Dbx1*-derived neurons, however, all rhythmic respiratory outputs were eliminated. The absence of both cervical and lumbar rhythms in *Dbx1* mutant mice suggests that in addition to its inspiratory rhythmogenic role in the preBötC, *Dbx1* is essential for the development of neurons necessary for the expression of rhythmic respiratory output from other motor pools, although the exact location of these neurons remains unknown. *Dbx1*-dependent neurons have recently been shown to play an important role in locomotor pattern generation (*Talpalar et al., 2013*).

The temporal overlap of lumbar and cervical bursts raised the question whether the lumbar output seen in E18.5 wild-type preparations is generated by the same networks that produce active expiration in more mature in vitro preparations or in vivo. We propose that this is the case for three reasons. First, we found that lumbar and cervical activities were not identical. Either output could occur either in isolation or with a distinct temporal separation. This same pattern of independent motor activity was also seen in respiratory output from internal intercostal muscles or VII motor roots, suggesting it is a general property of respiratory motor pools. Second, output from lumbar and thoracic roots (which innervate expiratory musculature) has previously been shown to require structures outside of the preBötC and the rostral ventral respiratory group, which are the well-established sources of rhythmic inspiratory drive and premotor innervation to the phrenic motor pool (*Miller et al., 1985*; *Merrill and Lipski, 1987*; *Sasaki et al., 1991*; *Iscoe, 1998*; *Janczewski et al., 2002*; *de Almeida et al., 2010*; *Ford and Kirkwood, 2013*). These anatomical differences are consistent with independent sources of respiratory drive to different motor pools serving inspiration and expiration, respectively, although we cannot rule out changes in connectivity during maturation (*Iscoe, 1998*). Third, simultaneous co-activation of cervical and lumbar roots is also present in older in vitro rat preparations, suggesting it is only the relative percentage of respiratory bursts showing one or the other pattern that changes, not the ability to generate each pattern (*Taccola et al., 2007*).

We further provide evidence that each respiratory motor pool receives input from independent but coupled oscillators that interact within the timescale of an individual respiratory event (*Carroll and Ramirez, 2013*; *Moore et al., 2013*). Respiratory networks are capable of reconfiguring to produce

different patterns of respiratory outputs, such as sighs, depending upon the behavioral or modulatory state of the organism (*Lieske et al., 2000*; *Doi and Ramirez, 2008, 2010*; *Chapuis et al., 2014*). We extend those findings to include activation of lumbar, thoracic, and VII motor pools, which are normally, but not obligatorily, active during different phases of the respiratory cycle. We observed that sigh-like cervical patterns corresponded with lumbar outputs active primarily only during the initial eupneic phase, whereas intercostal and VII outputs were active during both biphasic cervical bursts. In contrast to these motor outputs, 95% of preBötC inspiratory neurons are active during both inspiratory bursts and during sighs, which suggests that the differential gating of motor output occurs outside the putative inspiratory oscillator or by transient reconfigurations within respiratory populations (*Tryba et al., 2008*; *Chapuis et al., 2014*). We speculate that the differences in the timing and presence of respiratory outputs from lumbar and thoracic motor pools between these late fetal and more mature preparations are related to differences in the balance of synaptic interactions between hindbrain neurons and not the absence of connectivity (*Thoby-Brisson et al., 2009*; *Bouvier et al., 2010*; *Perreault and Glover, 2013*).

The loss of both RTN and RL neurons does not preclude respiratory deletions but does lead to periods of transient decoupling of respiratory motor rhythms (e.g., *Figure 4H*). Between control and RL⁻/RTN⁻ mice, individual motor pools differed in their co-activation. Lumbar outputs increased but cervical decreased their percentage of solitary activity. These data are inconsistent with a role for RL and RTN populations in generating lumbar respiratory output. It is important to note that the strong coupling of the cervical, lumbar, and VII motor roots in RL⁻/RTN⁻ mice suggests that the excitatory connections likely arising from the preBötC are maintained (*Tan et al., 2010*; *Feldman et al., 2013*; *Moore et al., 2013*). In RL⁻/RTN⁻ mice, the temporal overlap between cervical and lumbar output was nearly complete, again raising the question of the independence of the drives to each motor pool. Because there is no evidence in the literature that loss of *Atoh1* would affect *Atoh1*-independent projection patterns, we suggest the temporal gap is lost because of differences in connectivity between hindbrain respiratory populations (*Miesegaes et al., 2009*; *Rose et al., 2009a*; *Kohl et al., 2012*; *Perreault and Glover, 2013*). We find that the loss of both RL and RTN neurons eliminates respiratory bursts with intervals longer than 150 ms between respiratory oscillators, that is, each population can be co-active or independent but lacks the capacity for the short temporal gaps necessary for normal breathing patterns.

We interpret these results to indicate that the loss of RL neurons eliminates a net inhibitory effect on preBötC *Dbx1* neurons, likely via reciprocally connected inhibitory interneurons. Recent work has found that biphasic inspiratory bursts (sighs) begin during late embryogenesis and that the temporal delay between eupneic and sigh components is dependent upon the strength of synaptic inhibition (*Chapuis et al., 2014*). Moreover, the density of the co-transporter necessary for setting the chloride equilibrium potential in the medulla increases significantly during the post-natal period (*Liu and Wong-Riley, 2012*). This suggests the difference in the relative percentage of respiratory cervical-lumbar bursts showing temporal delays above 150 ms between our E18.5 preparations and older animals may be simply a consequence of generally weaker synaptic inhibition (*Greer et al., 2006*).

We propose RL neurons play an essential role in the coordination of mammalian breathing behavior by modulating the inhibitory interneurons that produce the temporal lag between independent oscillators. This modulation also helps stabilize coupled oscillators preventing independent or unwanted activation. This has the advantage that temporal delays can be controlled without strongly affecting the rhythmogenic properties of the underlying oscillators (*Hill et al., 2003*; *Grillner, 2006*). It is also possible, however, that the loss of *Atoh1* neurons indirectly affects respiratory networks by modulating the development of inhibitory populations or the overall maturation of hindbrain circuits. It is important to point out that we were unable to selectively eliminate lumbar motor output in any of our genetic manipulations. While our data are consistent with the presence of independent respiratory oscillators, we cannot rule out that the variability we see between respiratory motor pools is the consequence of higher level network interactions between a single distributed respiratory oscillator and small pools of respiratory premotor neurons (*Smith et al., 2007*). Similarly, whether there are discrete anatomical boundaries between these putative independent oscillators is also unknown.

*Atoh1* is essential for the development of a number of important neural populations in vertebrates. One general characteristic of many of these populations is their importance for the temporal coordination of the activity of different populations, be it fast motor coordination for neurons of the cerebellum or auditory processing in the hindbrain (*Wang et al., 2005*; *Maricich et al., 2009*; *Miesegaes*

*et al., 2009*; *Rose et al., 2009a*). This temporal functionality is also consistent with previous data indicating a role for the *Atoh1*-derived RL neurons of the parabrachial nucleus in the phase-dependent inhibition of inspiratory output (*Richter et al., 1992*; *Gray, 2008*; *Rose et al., 2009a*). These data emphasize that phasic inhibition influences respiratory patterns but is not rhythmogenic per se, as has been suggested (*Smith et al., 2007*; *Abdala et al., 2009*; *Richter and Smith, 2014*).

An influential model proposes that simple behaviors are generated by glutamatergic interneurons controlling the activation of segmentally organized unit burst oscillators (central pattern generators) (*Grillner, 2011*). These excitatory unit-burst-oscillator networks are putatively coupled to inhibitory interneurons such that independent modules generate the activity underlying muscle groups that must be properly coordinated, for example, flexors and extensors across hinge joints in locomotion or specific left-right axial muscles in body segments during swimming. Recent work provides evidence for unit burst oscillator modules coordinated by inhibition in the mammalian spinal cord (*Dougherty et al., 2013*; *Hägglund et al., 2013*; *Talpalar et al., 2013*). We propose that the mammalian respiratory network is organized in a similar fashion, with separate oscillators driving specific cervical and lumbar motor pools, in which the preBötC plays the central organizing role (*Feldman et al., 2013*; *Moore et al., 2013*). The fact that RL glutamatergic neurons are not necessary for rhythmogenesis implies that the generation of rhythm and the modulation of phase timing are controlled by separate populations. This shared control is similar to proposed networks of coupled independent oscillators in invertebrate motor neurons but represents a novel viewpoint in vertebrate systems (*Hill et al., 2003*; *Jezzini et al., 2004*; *Mulloney and Hall, 2007*). Whether *Atoh1*-dependent neurons control temporal intervals between independent oscillators only in breathing or also in other rhythmic behaviors is an interesting question for future research.

## Material and methods

### Mice

Experiments were done in accordance with the Institute for Laboratory Animal Research Guide for the Care and Use of Laboratory Animals (*Care et al., 1985*). All experiments were approved by the Animal Studies Committee at Washington University School of Medicine, the Institutional Animal Care and Use Committee at The College of William & Mary, and the Center for Comparative Medicine, Baylor College of Medicine. We utilized *Atoh1*-Cre transgenic (*Atoh1*-Cre$^{Tg}$) (*Matei et al., 2005*), *Atoh1*$^{Cre/+}$ (*Yang et al., 2010*), *Atoh1*$^{LacZ/+}$ (*Ben-Arie et al., 1997*), *Atoh1*$^{Flox/Flox}$ (*Atoh1*$^{F/F}$) (*Huang et al., 2012*), *Dbx1*$^{LacZ/+}$ (*Pierani et al., 2001*), *Phox2b*-Cre (*Rossi et al., 2011*); *Rosa26*-stop-eYFP (R26;YFP) (*Srinivas et al., 2001*), *Rosa26*-stop-TD Tomato (*Madisen et al., 2010*), and *VGlut2*$^{Flox/Flox}$ (*VGlut2*$^{F/F}$) (*Tong et al., 2007*) mice. Mice were crossed and bred on a C57BL6 or mixed CD1/C57BL6 background.

### *En bloc* electrophysiology

Brainstem-spinal cord (*en bloc*) preparations with an anterior transection near the diencephalon–midbrain junction were made using E18.5 embryos delivered under anesthesia (ketamine/xylazine mixture) by cesarean section from timed-pregnant female mice. The dissections were done while keeping the embryos submerged in cold (4°C) artificial cerebral spinal fluid (regular/enhanced = WT [n = 17/5]; RL$^-$/RTN$^-$[n = 13/6]; RL$^+$/RTN$^-$[n = 5/1]; RL$^+$/RTN$^{silent}$[n = 3/0]; aCSF (in mM): 124 NaCl, 3/5 KCl, 1.5/2.4 CaCl$_2$, 1.0/1.3 MgSO$_4$, 25.0/26.0 NaHCO$_3$, 0.5 NaH$_2$PO$_4$/1.2 KH$_2$PO$_4$, 30 D-Glucose (Sigma, St. Louis, MO) equilibrated with 95% O$_2$ and 5% CO$_2$ to pH = 7.4) and transferred into a partitioned 6 ml recording chamber, which were separately gravity fed by reservoirs of heated (25°C–26°C) and aerated (95% O$_2$ and 5% CO$_2$) aCSF at a rate of 3–4 ml/min. After transferring the preparation into the recording chamber, the brainstem and spinal cord compartments were rendered mutually impervious with petroleum jelly. Allowing ~30 min for stabilization, the extracellular electrophysiological recordings were made (acquisition rate 4 kHz) simultaneously from a cervical (C2–C6) and a lumbar (L1) ventral spinal motor root using suction electrodes, differentially amplified (low noise Grass Instruments, band pass filtering [0.3–3 kHz]), digitized using an analog to digital converter (AD instruments, Colorado Springs, CO) and then integrated over time (absolute value with a 100 ms decay time constant) using LabChart 7 Pro software (version 7.2.4, AD Instruments). Similarly, triple recordings were also made using an additional suction electrode to record from facial nerve or a sharp tungsten microelectrode (10 mΩ impedance and 1 µm tip diameter; FHC, inc.) to record electromyograph from intercostal muscle XI (IC) in a non-partitioned bath as previously described in neonatal rats (*Iizuka, 1999*).

After recording baseline activity, 1 µM substance P (SP) or $10^{-12}$ M somatostatin (SST) in aCSF was added selectively to the brainstem compartment in the partitioned bath preparations. In some early experiments, we used an enhanced aCSF suggested to increase the percentage of respiratory bursts with lumbar activity (*Iizuka, 2004*; *Ruangkittisakul et al., 2008*). We found no changes in the percent of lumbar bursts nor any differences in the phase distributions of fictive breaths, so we combined both solutions when comparing deletion and time lag. The peak time and amplitude, phase difference and inter-burst intervals (period) were determined for cervical, lumbar, and facial nerve bursts as well as for IC. The comparisons of cervical and lumbar periods indicative of respiratory frequency were made between the different genotypes by direct comparison, whereas the amplitudes of the integrated respiratory bursts were first normalized against the baseline arithmetic mean determined individually. The percentage of inspiratory and expiratory deletions was also calculated from each recording and these values were then compared between treatments and mouse lines. In cases of triple recording involving facial nerve, the relative degree of phase separation for the lumbar and facial nerve bursts from the preceding cervical burst (Φ) was determined by equating the instantaneous cervical inter-burst interval period to 360°. Moreover, coupling efficiencies of rhythms whose integrated peaks occurred within 1 s period of a cervical inspiratory burst were quantified and compared in facial nerve and IC triple recordings.

## Slice recordings

Transverse slices (550 µm thick) from neonatal (P0-P2) *Atoh1*-Cre[TG] X Rosa26-stop-YFP or tdTomato mice were dissected and prepared for recordings as described previously (*Hayes and Del Negro, 2007*; *Picardo et al., 2013*). On-cell and whole-cell patch recordings were obtained using infrared-enhanced differential interference contrast videomicroscopy (IR-DIC) after fluorescent identification of *Atoh1*-derived neurons in both YFP and tdTomato reporter mice. ACSF contained (in mM): 124 NaCl, 9 KCl, 0.5 $NaH_2PO_4$, 25 $NaHCO_3$, 30 D-glucose, 1.5 $CaCl_2*2H_2O$, and 1 $MgSO_4$. Slices were placed into a 0.5-ml chamber within an upright fixed-stage microscope (Zeiss Microimaging, Thornwood, NY) and ACSF was perfused at ~5 ml/min at 27–28°C. Patch recordings employed a Dagan IX2-700 amplifier (Minneapolis, MN). Respiratory-related motor output was monitored from XII nerves with extracellular suction electrodes and a high-gain differential amplifier with band-pass filtering (0.3–1 kHz) (Dagan EX4-400), full-wave rectified and smoothed for display. Data were acquired using Chart software and a Powerlab 4/30 (ADInstruments). A liquid junction potential, which measured 1 mV, was not corrected in current-clamp experiments. We used the following patch solution containing (in mM): 140 K-gluconate, 5 NaCl, 0.1 EGTA, 10 HEPES, 2 Mg-ATP, and 0.3 $Na_{(3)}$-GTP (pH = 7.2 using KOH). We added 2–4 µl/ml of Alexa Fluor 568 hydrazide ($Na^+$ salt, Invitrogen, Carlsbad, CA) to the patch solution for fluorescent visualization of morphology. Neurons were visually identified using both IR-DIC videomicroscopy and epifluorescence illumination (X-Cite 120, EXFO, Montreal, Canada) and a fluorescent filter to identify YFP or tdTomato labeled cells. Images were acquired of recorded *Atoh1* neurons.

## Immunohistochemistry

Tissue sections were washed in PBS with 0.2% triton X-100, blocked in 10% heat inactivated normal horse sera, incubated in antibody overnight at 4°C, incubated in secondary antibody and coverslipped in Vectashield or Prolong Gold.

## Antibodies

Chicken anti-beta galactosidase (LacZ) 1:4000 (Abcam, Cambridge, MA), Chicken anti-green fluorescent protein (GFP) 1:1000 (Aves Labs, Tilgard, OR), Lbx1 (1:10,000, gift from C Birchmeier), Rabbit anti-neurokinin 1 receptor (NK1R) 1:2000 (Millipore, Billerica, MA), Goat anti-Phox2b 1:500 (Santa Cruz Biotechnology (SCBT), Santa Cruz, CA), Goat anti-Islet-1 1:500, (Neuromics). All antibodies used have been previously characterized and no signals were present in genetic or antibody controls.

## In situ hybridization

As previously described (*Gray, 2013*), slides were immersed in 4% PFA, permeabilized with proteinase K or RIPA Buffer, washed in 0.1 M triethanolamine-HCl with 0.25% acetic anhydride, blocked in hybridization buffer at 65°C, then placed into slide mailers containing hybridization buffer with DIG-labeled antisense RNA at 1 µg/ml overnight at 65°C. Slides were washed in SSC buffers at 62°C, then washed and incubated in alkaline phosphatase conjugated anti-DIG antibody in 10%

NHS and incubated in NBT-BCIP until cellular labeling is clear. For combined immunohistochemistry and in situ hybridization, slides are stained for mRNA expression prior to immunohistochemical labeling.

## Genotyping

Mice were genotyped by PCR using primers specific for $Atoh1^F$, $Atoh1^{LacZ}$, Cre recombinase, $Dbx1^{LacZ}$, GFP/YFP, TD-Tomato, and $VGlut2^F$, as previously described (**Ben-Arie et al., 1997**; **Pierani et al., 2001**; **Srinivas et al., 2001**; **Matei et al., 2005**; **Tong et al., 2007**; **Madisen et al., 2010**; **Yang et al., 2010**; **Rossi et al., 2011**; **Huang et al., 2012**).

## Tissue acquisition

Neonatal pups (P0–P4) or embryos from timed pregnant females (morning of plug = E0.5, E18.5) were anesthetized and perfused with 4% paraformaldhyde in 0.1 M PBS, pH 7.4. Embryos or isolated brainstems were postfixed in PFA overnight at 4°C, cryoprotected in 25% sucrose in PBS, blocked, frozen in OCT, and stored at −75°C. Brainstems were sectioned in sets of six on a Hacker (Winnsboro, SC) cryostat at 20 µm and sections are thaw mounted onto Superfrost Plus slides and stored at −20°C until use.

## IHC and ISH image acquisition

Fluorescent and brightfield images were acquired using a Nikon 90i microscope (Nikon Instruments, Melville, NY), Roper H2 cooled CCD camera (Photometrics, Tucson, AZ), and Optigrid Structured Illumination Confocal with a Prior (Rockland, MA) motorized translation stage. Pseudo-colored images were acquired in Velocity (Perkin Elmer, Waltham, MA), and modified in Photoshop (Adobe, San Jose, CA) and exported as 8 bit JPEG images. Images were filtered and levels were modified for clarity.

## Temporal delay and phase histograms

Temporal interval values (τ, in ms) and respiratory phase (Φ, in °) were binned (delay–12 ms, phase–5°), and graphed in Igor Pro (Wavemetrics, Lake Osego OR) then exported to Adobe Illustrator and scaled to a standard height for comparison. Delay histograms were generated from 1088 (WT Baseline), 789 (WT SP), 373 (WT SST), 1522 ($Atoh1^{LacZ/LacZ}$ baseline), 318 ($Atoh1^{LacZ/LacZ}$ SP), and 1047 ($Atoh1^{LacZ/LacZ}$ SST) respiratory-related bursts. Phase histograms were generated from WT cervical-lumbar or cervical-VII breaths (506, 500) and $Atoh1^{LacZ/LacZ}$ cervical-lumbar or cervical-VII bursts (2125, 857).

## Statistical analysis

Respiratory periods, the distribution of time lag between inspiratory and expiratory burst peak times, the effect of time lag on the amplitude of respiratory bursts and percentage of respiratory deletions were compared using mixed random effects ANOVA with mouse as the random effect and least-squares mean values were determined. The percentage of fictive breaths with respiratory phase lag were compared between WTs and $RL^-$/$RTN^-$ mice using independent samples $t$ test and One-Way ANOVA followed by Tukey's honestly significant difference test for the effect of peptides with-in these genotypes (*p<0.05, **p<0.01, #p<0.001).

## Acknowledgements

The authors thank Dr Michael J Wallendorf of the Washington University School of Medicine Division of Biostatistics for his assistance with statistical analysis. We are grateful to Bradford Lowell and Joel Elmquist for sharing $SLC17A6$ ($VGlut2$)$^{F/F}$ and $Phox2b$-Cre mice, respectively. This work was supported by American Heart Association SouthWest affiliate Predoctoral Fellowship to WHH (11PRE6080004); Howard Hughes Medical Institute to HYZ; National Heart, Lung, and Blood Institute (NHLBI) Grant to CDN (R01-HL104127); National Institute of Neurological Disorders and Stroke, Ruth L Kirchstein National Research Service Award Fellowship Grant to MCDP (F31-NS071860); and NHLBI Grant to ST and PAG (R01HL089742).

## Additional information

### Competing interests

HYZ: Senior editor, *eLife*. The other authors declare that no competing interests exist.

## Funding

| Funder | Grant reference number | Author |
|---|---|---|
| National Institutes of Health | R01HL089742 | Srinivasan Tupal, Guang-Yi Ling, Paul A Gray |
| Howard Hughes Medical Institute | | Wei-Hsiang Huang, Huda Y Zoghbi |
| American Heart Association | 11PRE6080004 | Wei-Hsiang Huang |
| National Institutes of Health | R01HL104127 | Maria Cristina D Picardo, Christopher A Del Negro |
| National Institutes of Health | F31 NW071860 | Maria Cristina D Picardo |

The funders had no role in study design, data collection and interpretation, or the decision to submit the work for publication.

## Author contributions

ST, W-HH, CADN, PAG, Conception and design, Acquisition of data, Analysis and interpretation of data, Drafting or revising the article; MCDP, Acquisition of data, Analysis and interpretation of data; G-YL, Acquisition of data, Contributed unpublished essential data or reagents; HYZ, Conception and design, Drafting or revising the article

## Author ORCIDs

Maria Cristina D Picardo, http://orcid.org/0000-0001-8912-2175
Christopher A Del Negro, http://orcid.org/0000-0002-7848-8224

## Ethics

Animal experimentation: Experiments were done in accordance with the Institute for Laboratory Animal Research Guide for the Care and Use of Laboratory Animals. All experiments were approved by the Animal Studies Committee at Washington University School of Medicine (protocol # 20110249), the Institutional Animal Care and Use Committee at the College of William and Mary, and the Center for Comparative Medicine, Baylor College of Medicine.

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
