## [Decision Letter]

[Editors’ note: a previous version of this study was rejected after peer review, but the authors submitted for reconsideration. The two decision letters after peer review are shown below.]

Thank you for choosing to send your work entitled “*Atoh1*-dependent, rhombic lip neurons are required for inspiratory-expiratory temporal coordination” for consideration at *eLife*. Your full submission has been evaluated by a Senior editor and 3 peer reviewers, one of whom is a member of our Board of Reviewing Editors, and the decision was reached after discussions between the reviewers. We regret to inform you that your work will not be considered for publication. *eLife* has a policy to not ask for revisions that would require considerable new data collection. As you will see below, the reviewers believe that new experiments are likely to be necessary, and therefore the decision to reject was taken to allow you to move onto another journal if you wish to do so. On the other hand, as you will see below, the reviewers found your work potentially of great value, so if you believe that you can answer the substantial and substantive objections raised in the review, we would consider a new submission for review.

The following individuals responsible for the peer review of your submission have agreed to reveal their identity: Ronald Calabrese (reviewing editor) and Nino Ramirez (peer reviewer).

The authors combine genetic and electrophysiological experimental studies to address the role of RL and RTN neurons in the control of respiration in the mouse brainstem. Using genetic lines that eliminate neurons that require certain transcription factors they assess the role of these genetic deletions on the breathing rhythm assayed electrophysiologically in the embryonic brainstem/spinal cord. They provide evidence that RL neurons do not generate the expiratory rhythm but are necessary for the coordination of expiration and inspiration. They further explore the origins of the expiratory rhythm itself. Their hypotheses about how expiration is generated and about the role of RL neurons in coordinating expiration and inspiration are novel and potentially transformative in the field. All three reviewers had high enthusiasm for these ideas but serious concerns that the authors do themselves and the field a disservice by publishing prematurely without necessary supporting data and analyses. The expert reviewers have several concerns detailed in their reviews that should be helpful to the authors.

Three major issues should be addressed to substantiate the conclusions of the authors so that these conclusions can be widely accepted.

1) The authors must provide supporting evidence that their expiratory recordings indeed show expiratory bursts and not inspiratory or mixed bursts.

2) The authors must precisely and quantitatively define coordination between expiration and inspiration. It is very confusing that the 'proper coordination' appears sporadic even in control preparations - moreover synchrony is a coordination.

3) The authors must clearly show that what they call sighs are indeed sighs and describe in detail their internal coordination.

The various expert reviewers address these 3 issues from different vantage points but all agree on their importance.

[Editors’ note: minor comments have not been included in the reviews below.]

Reviewer #1:

The authors combine genetic and electrophysiological experimental studies to address the role of RL and RTN neurons in the mouse brainstem. Using genetic lines that eliminate neurons that require certain transcription factors they assess the role of these genetic deletions on the breathing rhythm assayed electrophysiologically in the embryonic brainstem/spinal cord with special reference to the expiratory rhythm which is currently not well understood. They find that the RL and RTN are not necessary or sufficient for the expiratory rhythm and uncover a hidden inhibitory role of the RL on the PreBotC, which controls the inspiratory rhythm. They then go on to show that the RL is necessary for expiration to lead inspiration as is the 'predominant' mode in adults and conclude it provides coordinating inhibition to the inspiratory rhythm generator in the PreBotC. Their results further indicate an independent but unidentified expiratory rhythm generator that requires the Dbx1 TF.

The writing is succinct but the paper is very dense and not that easy to follow. The figures are very complex and the legends hyperdense, but do provide the necessary data. The work appears carefully done with sufficient attention to detail and adequate numbers of experiments. The work may prove controversial in the respiration community because does not fit with existing conclusion about the role of RL and RTN in expiration, but really the results are hard to assail on that front. The respiratory rhythm generator is a well-studied motor network in vertebrates and has great significance for understanding general mechanisms in the brain and in health.

Concerns:

1) I am not qualified to judge the genetic deletions for completeness and specificity but there is supporting data that appears sound to a non-expert.

2) My major problem with the paper arises from the loose manner in which the coordination between inspiration and expiration is described. Originally it is claimed that a lag of >150 ms between expiration and inspiration is normal for the adult nervous system “...active expiration usually precedes inspiratory output by hundreds of milliseconds (when both are present...'), but the % of fictive 'breaths' with this relation is not given. Then the normal WT coordination in the embryonic nervous system is described with only ∼26% of fictive 'breaths' show a lag of >150 ms between expiration and inspiration. Thus it is unclear to me what is normal for this coordination: near synchrony or an expiratory lead (inspiratory lag). If majority %ages of an expiratory lead (inspiratory lag) are seen in adult nervous system then adult-like is appropriate as a description for this coordination. Even then the authors should each time make clear that they are speaking of coordination at all times, because the inspiration and expiration can be near coincident (i.e., near synchrony is a coordination) but that the ability of expiration to lead is missing in the transgenic lines (genetic deletions). Thus coordination is always present but certain intervals, rare though they may be in WT embryonic nervous system, are not present in the transgenics. Making this distinction clear is important because otherwise it is hard for a non-respiration expert to follow. In several places in the text I have high-lighted places where the authors should make this distinction.

Reviewer #2:

Despite the fact that the paper proposes an interesting new hypothesis there are several concerns that need to be addressed in order first to make the figures more supportive to the conclusions made (in other terms given the illustrations presented the authors are over-interpreting their data), to render the paper more demonstrative (average and quantifications required) and the writing more rigorous. Amongst the large number of concerns only the major ones are detailed below:

- Two recent studies demonstrated that Dbx1-derived neurons do not contribute to the RTN/pFRG oscillator (4, 21). The authors should justify why they consider some Dbx1-derived neurons as part of the RTN/pFRG in the introduction?

- Why the expiratory bursts recorded from L1 do not show a typical bi-phasic discharge pattern as demonstrated by [41]?

- How do the authors explain that Substance P has no effect on the inspiratory period in the WT?, and that SP increases inspiratory period in the RL-/RTN- animals?

- The entire demonstration that RL neurons exert an inhibitory effect on the respiratory networks is absolutely not convincing. Sigh definition, samples selected for the illustration, the lack of quantification and contradictions in different part of the text prevent this part of the study to be demonstrative. It has been previously shown that inhibition serves to temporally associate the two components of a sigh burst (46). Are the mutants generating two types of burst differing in terms of amplitude and frequency with no time relation (as it is the case in WT after blockade of inhibition)? If yes then the possibility that inhibition is deficient could eventually be proposed, but otherwise the complete absence of sigh does not necessarily mean a deficit in inhibition. In addition the sigh generating mechanisms have been shown to be an intrinsic properties of the preBötzinger network itself (46; 99), so it is unclear how an inhibition originating from the RL neurons would play any role.

- The observation that L1 activity is missing in the Dbx mutant is interesting because this has not been documented yet. However, the conclusion that the expiratory oscillator might depend upon Dbx1-derived neurons might be an overstatement. Indeed Dbx1 could be required for the specification of one element of the expiratory pathway but not necessarily the oscillator per se.

- The authors should justify why they chose to record from YFP-Atoh1 derived neurons located in the ventrolateral medulla rostral to the preBötC but caudal to the RTN and not in other Atoh1-positive neuronal populations.

One example of a YFP+ neuron receiving inhibition during inspiration (called expiratory neuron in the legend) is presented. Is this neuron part of the expiratory generator? If yes then we can suspect that Dbx1-derived neurons are also Atoh1+. Does such a population of neurons exist in the Bötzinger region? The authors used this neuron as an indication that at least some Atoh1 neurons within the medulla are synchronized to respiratory output. What can be the relevance of a result observed only once??

- Figures 4 and 5: a single sample used as an illustration without any average traces, quantification of any parameters and long term recordings is not a strong demonstration that could support any conclusions. Also the samples used to illustrate sigh-like bursts are not typical and the number of bursts used to build the lag plots should be provided.

Reviewer #3:

This is a very important study by Tupal, Huang et al. and Paul Gray as senior author.

This study has the potential to become an important conceptual and methodological contribution to the field, as it offers new ideas how inspiration and expiration are generated by the respiratory network. It is “generally” thought that inspiration and expiration are generated by two independent, yet interacting oscillators. One possibility is that the AtOH1 neurons generate expiration while the DBX1 neurons generate inspiration. Using elegant transgenic approaches the authors demonstrate that DBX1 neurons generate both respiratory phases and that the AtOH1 neurons establish the delay between these phases. This is a fundamental departure of the current concepts, which makes this study very interesting. The model proposed in this study is well explained. However, there are three main caveats that need to be addressed:

1) The activity characterized in this study represents an embryonic rhythm. While the authors argue that the activity is similar to a mature network activity, I am not convinced that the data presented are sufficiently strong to arrive at this conclusion.

2) The identification of the respiratory phases is based on root recordings and not on actual EMG activity from inspiratory and expiratory muscles. This is a major comment, because the entire study hinges on the proper identification of inspiratory and expiratory activity.

3) The identification of a sigh burst is vague, which might lead to a misleading conclusion.

These caveats could be addressed experimentally – by performing additional experiments, or by adding clearly stated caveats into the discussion.

[Editors’ note: what now follows is the decision letter after the authors submitted for further consideration.]

Thank you for sending your work entitled “*Atoh1*-dependent rhombic lip neurons are required for temporal delay between independent respiratory oscillators” for consideration at *eLife*. Your article has been favorably evaluated by a Senior editor and 3 reviewers, one of whom is a member of our Board of Reviewing Editors.

The following individuals responsible for the peer review of your submission have agreed to reveal their identity: Ronald Calabrese (Reviewing editor); Nino Ramirez (peer reviewer).

The Reviewing editor and the other reviewers discussed their comments before we reached this decision, and the Reviewing editor has assembled the following comments to help you prepare a revised submission.

The authors combine genetic and electrophysiological experimental studies to address the role of RL and RTN neurons in the control of respiration in the mouse brainstem. Using genetic lines that eliminate neurons that require certain transcription factors they assess the role of these genetic deletions on the breathing rhythm assayed electrophysiologically in the embryonic brainstem/spinal cord. They provide evidence that neither RL Atoh1 neurons nor RTN Phox 2b neurons generate the expiratory rhythm but RL Atoh1 neurons tune the sequential activation of independent oscillators essential for the fine control of different muscles during breathing. Their hypotheses about how expiration is generated and about the role or RL neurons in coordinating expiration and inspiration are novel and potentially transformative in the field. The paper contains new data and analyses that improve it over a previous submission.

1) The manuscript needs revision to make the presentation more accessible to a general audience. The data and figures and indeed the writing describing the data/figures are very complicated and the general reader will need a lot more guidance. Even experts in respiration need more guidance. Perhaps one method that will help the reader is to have the proposed circuit of Figure 8 brought up front in Figure 1 as an hypothesis to be tested and at the same time introduce the icons of Figure 8. Then for each set of traces these icons could be used to guide the reader to see what interaction is being illustrated. The writing needs to use these iconic guides. The icons will also help is showing which neuron group is genetically eliminated in each experiment and/or is the target of peptide modulation or drugs and how these manipulations adjust the proposed circuit.

Creative use of icons and adjustments in writing could make the paper a lot easier to follow. Any way that the figures can be simplified by reducing the number of individual traces would be helpful. The legends to the figures need to be made as clear as possible even if this means expanding the number of words, and they should refer to the icons in each case. We found the delay histograms of Figure 6 and the phase histograms of Figure 7 particularly difficult because they are described by overly dense and telegraphic legends. The supplemental figures need to be incorporated into the flow directly or eliminated.

2) The analysis of “sighs” remains a problem from the last submission. The authors should simply refer to these as double bursts in the Results and then in Discussion bring up their possible relationship to sighs. The double bursts should be quantified and a conservative course set in Discussion.

3) Another concern from the original submission is that only ∼25% of the normal bursts show 'normal' coordination. There should be some comment on this issue. How do you interpret this low percentage? It would seem to imply that the network capriciously teeters between Figure 8 and Figure 8. Is this how you see it?

4) The question of independent oscillators is complicated. It is possible to view the results as supporting 3 independent oscillators. On the other hand, as another reviewer points out, all the oscillations are driven by DBX1 cells and differential sensitivity to opiates has not been established so independent oscillators may not be supported. The authors need to address this issue squarely.

---

## [Author Response]

[Editors’ note: the author responses to the first round of peer review follow.]

*1) The authors must provide supporting evidence that their expiratory recordings indeed show expiratory bursts and not inspiratory or mixed bursts*.

*The reviewers wondered whether rhythmic output from lumbar nerve roots was directly related to expiration as seen in vivo because, in late embryonic preparations, the peak of cervical output (inspiration) and the peak of lumbar output (possibly expiration) overlap temporally the majority of the time. This is an important question, because if what we are recording actually represents inspiration then our data say nothing about the genetic origin of expiratory motor output*.

First, we recognize that the use of the term expiration to describe motor output, which (at best) occurs slightly before inspiration and more often is co-active with inspiration, is confusing. Expiration itself is often divided into stages with the period just prior to inspiration being called either late expiration or pre-inspiration. Even inspiration itself is more complicated, as each specific burst could be a eupneic breath, a sigh, a gasp or unrelated to breathing. We have decided it is more precise to step back from the terms inspiration and expiration but rather to simply name the motor root we are recording from, i.e., cervical or lumbar. We recognize it is widely accepted to describe cervical motor output as inspiration, but using only that one term was somewhat unwieldy. This is similar to descriptions of motor output in invertebrate systems. We do in places use the term “breath” in relation to the motor bursts to reinforce the respiratory-related nature of these outputs.

To address whether lumbar motor nerve roots generate expiratory-related output in more mature animals, we performed two different sets of new experiments. First, at the suggestion of all three reviewers, we examined the relationship between cervical and lumbar root activity with EMG muscle activity recorded from the XIth internal intercostal space (ICXI) in wild-type animals (Figure 2). ICXI was chosen because it was accessible, allowed for simultaneous cervical and lumbar root recording, and, most importantly, was previously shown to be active out of phase with the inspiratory drive from cervical roots both in vivo and in previous in vitro experiments (34). ICXI is driven by motoneurons in the caudal thoracic spinal cord, so it sits between the cervical and lumbar motor pools previously analyzed. Consistent with our previous recordings, the peak of ICXI activity most often occurred close in time to both cervical and lumbar respiratory outputs. However, also similar to previous recordings, the output of ICXI was not identical to cervical or lumbar outputs. Even in breaths in which lumbar output preceded cervical output, ICXI could be co-active with lumbar or both lumbar and cervical (Figure 2). In contrast, in breaths with biphasic sigh-like cervical bursts, ICXI was most active during the sigh-like burst (Figure 5). Thus when recording from 3 roots, we found the timing and presence of activity were not consistent with being inspiratory output. These data are also consistent with previous work in rats and cats showing that respiratory drives to thoracic and lumbar motor pools, which drive caudal intercostal and abdominal muscles, emanate from neurons outside the preBötzinger Complex or rVRG, the proposed sources of inspiratory rhythm generation and phrenic premotor drive, respectively.

From each litter of Atoh1LacZ/+ crosses we have a window of a few minutes after C-section to guess which animal has the correct genotype based purely on the absence of breathing. This extremely limited time window and guesswork, coupled with the difficulty of the EMG recording experiments per se, has precluded successful EMG recordings in Atoh1 mutant mice. For many reasons, including both resource limits and ethical concerns about the number of animals that will need to be bred and sacrificed to generate a reasonable number of these recordings, we have decided not to continue these experiments.

Because the question of the relationship of lumbar output to expiration is important, we performed an additional series of new in vitro experiments in which we simultaneously recorded from lumbar, cervical, and VII motor roots in both WT and Atoh1 mutant animals (new Figure 4). Previous work has shown that VII output can occur in preparations that lack the preBötC. Also, The VII nucleus is directly adjacent to population(s) proposed to constitute a preBötC-independent respiratory oscillators related to expiration. In WT we again found that many breaths corresponded to overlapping co-activation of cervical, lumbar, and VII roots. Additionally, we found that each root can be active independently of the other and that for any breath any temporal sequence of the three motor roots is possible. In Atoh1 mutant mice, this same pattern of co-expression and independence of motor pool output persists. Importantly, however, we found that the pattern of independent rhythmic output we noticed in cervical-lumbar recording was more pronounced when analyzing three simultaneous root recordings.

We interpret these data to mean that in WT E18.5 mice, the motor outputs we record from lumbar, thoracic, and VII roots are driven by different neuron populations of neurons than the cervical inspiratory oscillator. We also suggest that these independent networks persist and play an essential role in the generation of cervical, lumbar, and VII motor output from Atoh1 mutant mice. We are clear to say that these different networks interact and their interaction is responsible for the temporal delays and coordination we see between motor pools. This is similar to recent findings in spinal cord networks showing independent activation of flexor and extensor motor outputs (26)

*2) The authors must precisely and quantitatively define coordination between expiration and inspiration. It is very confusing that the 'proper coordination' appears sporadic even in control preparations - moreover synchrony is a coordination*.

The reviewers are correct that co-activation is coordination. We replaced the term ‘coordination’ with ‘temporal delay’ when describing our results throughout the revised manuscript including the Title and Abstract.

*3) The authors must clearly show that what they call sighs are indeed sighs and describe in detail their internal coordination*.

The reviewers are correct that co-activation is coordination. We replaced the term ‘coordination’ with ‘temporal delay’ when describing our results throughout the revised manuscript including the title and abstract.

*The various expert reviewers address these 3 issues from different vantage points but all agree on their importance*.

*[Editors’ note: minor comments have not been included in the reviews below*.*]*

Reviewer #1:

*The authors combine genetic and electrophysiological experimental studies to address the role of RL and RTN neurons in the mouse brainstem. Using genetic lines that eliminate neurons that require certain transcription factors they assess the role of these genetic deletions on the breathing rhythm assayed electrophysiologically in the embryonic brainstem/spinal cord with special reference to the expiratory rhythm which is currently not well understood. They find that the RL and RTN are not necessary or sufficient for the expiratory rhythm and uncover a hidden inhibitory role of the RL on the PreBotC, which controls the inspiratory rhythm. They then go on to show that the RL is necessary for expiration to lead inspiration as is the 'predominant' mode in adults and conclude it provides coordinating inhibition to the inspiratory rhythm generator in the PreBotC. Their results further indicate an independent but unidentified expiratory rhythm generator that requires the Dbx1 TF*.

*The writing is succinct but the paper is very dense and not that easy to follow. The figures are very complex and the legends hyperdense, but do provide the necessary data. The work appears carefully done with sufficient attention to detail and adequate numbers of experiments. The work may prove controversial in the respiration community because does not fit with existing conclusion about the role of RL and RTN in expiration, but really the results are hard to assail on that front. The respiratory rhythm generator is a well-studied motor network in vertebrates and has great significance for understanding general mechanisms in the brain and in health*.

*Concerns*:

*1) I am not qualified to judge the genetic deletions for completeness and specificity but there is supporting data that appears sound to a non-expert*.

The completeness of genetic deletion is always of concern. We, and others, have previously published detailed analysis of the mice used in these experiments. The anatomical losses of both the Atoh1LacZ/LacZ and Phox2b-Cre;Atoh1LacZ/F within the RTN region have been published (Rose et al., 2009; [29]). Both the Phox2b-Cre and the VGlut2F/F have been extensively characterized and were used as control for specificity of the other lines (98; 80).

*2) My major problem with the paper arises from the loose manner in which the coordination between inspiration and expiration is described. Originally it is claimed that a lag of >150 ms between expiration and inspiration is normal for the adult nervous system “...active expiration usually precedes inspiratory output by hundreds of milliseconds (when both are present...'), but the % of fictive 'breaths' with this relation is not given. Then the normal WT coordination in the embryonic nervous system is described with only ∼26% of fictive 'breaths' show a lag of >150 ms between expiration and inspiration. Thus it is unclear to me what is normal for this coordination: near synchrony or an expiratory lead (inspiratory lag). If majority %ages of an expiratory lead (inspiratory lag) are seen in adult nervous system then adult-like is appropriate as a description for this coordination*.

We inadequately explained how our findings relate to the existing literature in adults. We clarified the text throughout the revision. What is “normal” is complicated. The respiratory field has largely been interested in the question of inspiration and most (nearly all) of the analysis surrounding expiratory activity has been qualitative. Normally expiration is completely passive in adult mammals. Active respiratory output to abdominal and internal intercostal muscles is state-dependent and usually requires a stimulus such as CO2 or hypoxia, which boosts respiratory function overall. Motor output is then described as expiratory if it is not co-active with the diaphragm. Abdominal motor output is silent or at least decreased during inspiration. There is actually very little quantitative analysis of the temporal relation between late expiration (also referred to as pre-inspiration, E2, or Eb) with inspiration. We found one reference where we could infer the delay between abdominal and phrenic output to vary between less than 300ms to over 500 ms in an adult, anesthetized rat breathing elevated CO2 or decreased O2 (38). We also cite a second reference showing that a minority of post-natal in vitro preparations also produce breaths where cervical and lumbar roots are co-active (91). Together we interpret these data to indicate that the patterns of motor output we see in E18.5 WT preparations are relevant for understanding how adult networks produce expiratory behaviors.

*Even then the authors should each time make clear that they are speaking of coordination at all times, because the inspiration and expiration can be near coincident (i.e., near synchrony is a coordination) but that the ability of expiration to lead is missing in the transgenic lines (genetic deletions). Thus coordination is always present but certain intervals, rare though they may be in WT embryonic nervous system, are not present in the transgenics. Making this distinction clear is important because otherwise it is hard for a non-respiration expert to follow. In several places in the text I have high-lighted places where the authors should make this distinction*.

We modified the title, abstract, and text to be more specific. For example, in our previous abstract we stated “Without RL neurons, however, temporal coordination between independent oscillators is abolished.” This has been modified to “Without Atoh1 neurons, however, the requisite temporal delays between motor pools are abolished and coupling between independent respiratory oscillators decreases”. Similarly the section previously titled “Rhombic lip neurons are necessary for inspiratory-expiratory temporal coordination” is now titled “Rhombic lip neurons are necessary for respiratory motor pool temporal delay”. Within that section, the sentence which previously read “These data indicate these reduced preparations can produce adult-like patterns of respiratory output complete with tight temporal coordination,” now reads “These data indicate these reduced preparations can produce the adult-like respiratory pattern of lumbar activity preceding cervical activity” .

Reviewer #2:

*Despite the fact that the paper proposes an interesting new hypothesis there are several concerns that need to be addressed in order first to make the figures more supportive to the conclusions made (in other terms given the illustrations presented the authors are over-interpreting their data)*, *to render the paper more demonstrative (average and quantifications required) and the writing more rigorous. Amongst the large number of concerns only the major ones are detailed below:*

*- Two recent studies demonstrated that Dbx1-derived neurons do not contribute to the RTN/pFRG oscillator (*[4]*,*
[21]*)*. *The authors should justify why they consider some Dbx1-derived neurons as part of the RTN/pFRG in the introduction?*

The terminology of respiratory regions within the caudal pons is confusing even for many of us in the field. The term RTN/pFRG has been used to describe the general region of the brain thought to contain a putative expiratory oscillator but does not define a unique population nor does it have accepted boundaries (see (14; 18)).

The term RTN is accepted to represent a population of Atoh1+, Phox2b+, TH-, Lbx1+, VGlut2+ neurons surrounding the VII (25). In adult rats these neurons play an important role in chemosensitivity. This same population has also been called the ePF, embryonic parafacial region by Fortin and colleagues (96). In Dbx1 mutant mice the Phox2b+ RTN neurons are not affected.

The term pFRG (parafacial respiratory group) was defined functionally, ten years ago, by voltage-sensitive dye studies that identified a region near the VIIn that was active prior to the preBötzinger Complex (63). The region defined by these experiments only partially overlaps the Phox2b+ neurons of the RTN. While some neurons with firing patterns consistent with being pFRG neurons have been shown to express Phox2b, the genetic identity and boundaries of the pFRG are undefined. Other work in adult rats also suggests populations adjacent to the VII, but outside the RTN, strongly affects breathing and expiration.

Thus several groups have used the term RTN/pFRG as an imprecise but inclusive term to describe the general region surrounding the VII that plays a role in modulating breathing. I should point out that prior to the identification of specific genetic markers, the boundaries of the preBötC were similarly unclear between 1991 and 1999, and then it was ten more years before the developmental and genetic basis for understanding the preBötC core came to the fore.

*- Why the expiratory bursts recorded from L1 do not show a typical bi-phasic discharge pattern as demonstrated by*
[41]*?*

We describe this above in response to reviewer #1. Briefly, we do see breaths with lumbar and cervical delay, but they represent only a subset of breaths. Iizuka has shown that the phase of abdominal output changes between post-natal and adults (36). The differences we see likely relate to differences in the maturation of respiratory circuits.

*- How do the authors explain that Substance P has no effect on the inspiratory period in the WT?*, *and that SP increases inspiratory period in the RL-/RTN- animals?*

The effect of SP on breathing depends heavily on the model system and the baseline conditions. In anesthetized in vivo rat, SP injection directly within the preBötC does not affect respiratory frequency, but produces sighs (Gray 2001). In in vitro experiments, the SP only had effects on frequency in preparations with slow initial frequencies (Yamamoto et al., 1992). This is in contrast to a strong frequency effect of SP on preBötC slices. In our experiments we include the entire hindbrain and the neurokinin 1 receptor is expressed in many places outside the respiratory regions we have focused on here. This includes the Atoh1-derived dorso-lateral pons.

We used SP because it has become a fairly standard stimulant of breathing and useful for comparing between papers. We find that SP does not have a statistically significant effect on frequency in either wild type or Atoh1 mutant mice but does increase the very slow rhythm of RTN- mice. This, in part, may be due to the increased instability of these late embryonic preparations. As SP did not produce a change in the pattern of lumbar activation but did affect deletions and pattern formation, especially in Atoh1 mutant, we did not focus on analyzing the frequency effect.

*- The entire demonstration that RL neurons exert an inhibitory effect on the respiratory networks is absolutely not convincing*.

We agree this was an over-interpretation of the data.. We have modified the text in the discussion from “We interpret these results to indicate that RL neurons inhibit preBötC Dbx1 neurons, via reciprocally connected inhibitory interneurons” to We interpret these results to indicate that the loss of RL neurons eliminates a net inhibitory effect on preBötC Dbx1 neurons, likely via reciprocally connected inhibitory interneurons, and play an essential role in the coordination of mammalian breathing behavior by modulating the temporal lag between independent oscillators”. We are careful, however, to point out both in the initial description of our model and in the discussion that this inhibition in an interpretation of our results and not direct evidence of inhibition.

*Sigh definition, samples selected for the illustration, the lack of quantification and contradictions in different part of the text prevent this part of the study to be demonstrative. It has been previously shown that inhibition serves to temporally associate the two components of a sigh burst (*[46]*). Are the mutants generating two types of burst differing in terms of amplitude and frequency with no time relation (as it is the case in WT after blockade of inhibition)? If yes then the possibility that inhibition is deficient could eventually be proposed, but otherwise the complete absence of sigh does not necessarily mean a deficit in inhibition. In addition the sigh generating mechanisms have been shown to be an intrinsic properties of the preBötzinger network itself (*[46]*;*
[99]*), so it is unclear how an inhibition originating from the RL neurons would play any role*.

We agree our discussion of sighs was unclear and have significantly modified our Discussion to better fit the data (see above and responses to Reviewer #3).

The amplitudes and periods of respiratory output in these late embryonic preparations is more variable than in older animals, so we do not have the power to define sighs in the absence of a biphasic cervical burst. We do point out, however, that the patterns of output we see during sigh bursts in the WT – activation of cervical nerve roots but inhibition of lumbar nerve roots – do persist in Atoh1 mutants. The only deficit is the absence of biphasic cervical bursts. RL neurons could modulate sighs, even in slices, because the RL neurons are present throughout the medulla, including at the level of the preBötC (18).

*- The observation that L1 activity is missing in the Dbx mutant is interesting because this has not been documented yet. However, the conclusion that the expiratory oscillator might depend upon Dbx1-derived neurons might be an overstatement. Indeed Dbx1 could be required for the specification of one element of the expiratory pathway but not necessarily the oscillator per se*.

We whole-heartedly agree. In the text we now state clearly that Dbx1 is necessary for the expression of the behaviors, not that Dbx1 neurons generate the rhythm. For example:

“Dbx1-derived neurons, however, are essential for the expression of both cervical and lumbar respiratory behaviors.”

“The absence of both cervical and lumbar rhythms in Dbx1 mutant mice suggests that in addition to its inspiratory rhythmogenic role in the preBötC, Dbx1 is essential for the development of neurons necessary for the expression of rhythmic respiratory output from other motor pools, although the exact location of these neurons is, as yet, unknown.”

*- The authors should justify why they chose to record from YFP-Atoh1 derived neurons located in the ventrolateral medulla rostral to the preBötC but caudal to the RTN and not in other Atoh1-positive neuronal populations. One example of a YFP+ neuron receiving inhibition during inspiration (called expiratory neuron in the legend) is presented. Is this neuron part of the expiratory generator? If yes then we can suspect that Dbx1-derived neurons are also Atoh1+. Does such a population of neurons exist in the Bötzinger region? The authors used this neuron as an indication that at least some Atoh1 neurons within the medulla are synchronized to respiratory output*. *What can be the relevance of a result observed only once??*

We recorded at that level to examine whether Atoh1 neurons were strongly respiratory-modulated, like the majority of neurons within the ventrolateral medulla in preBötC-containing slices. More rostral populations are not accessible in visualized patch recordings. Atoh1 neurons release glutamate, but are not part of the preBötC because they are not Dbx1-derived. Like Dbx1, however, Atoh1-derived neurons form a column of neurons extending the length of the hindbrain. Furthermore, Atoh1 neurons are present at the level of the BötC, but because they are not glycinergic, Atoh1 neurons would not be considered part of the BötC per se. As we see no change in lumbar motor output in Atoh1 mutant mice, these neurons are not part of the expiratory oscillator. Although we only saw a single respiratory Atoh1 neuron, it is evidence these types of neurons exist and provides some of the first evidence that expiratory neurons at the level of the preBötC can be excitatory.

*-*
Figures 4 and 5*: a single sample used as an illustration without any average traces, quantification of any parameters and long term recordings is not a strong demonstration that could support any conclusions. Also the samples used to illustrate sigh-like bursts are not typical and the number of bursts used to build the lag plots should be provided*.

We greatly appreciate the recommendation to provide average traces for these graphs. The new Figure 7 and Figure 7–figure supplement 1 show average traces in WT and Atoh1 mutant mice. This new figure now clearly demonstrates the flexibility of the WT pattern and lack of temporal lag in the Atoh1 mutant. As to the images provided, they are typical of the outputs we see in our experiments, although we also now include motor output from internal intercostal and root output from the VII. We point out that sigh bursts in in vitro preparations are usually acquired from field potentials within the preBötC. In our case, we report motor root output and do not see as large an increase in amplitude for the sigh component. Whether this is due to an increase in both excitatory and inhibitory firing in the preBötC compared to just the excitatory component that would be visible via the motor output is unknown.

We have updated the methods to provide burst #s. As described above in our response to Reviewer #1, there are few data in the literature describing the temporal relationships between cervical and lumbar bursts. We quantitatively analyzed of lag between cervical and lumbar bursts as well as phase between cervical, lumbar, and facial bursts.

Reviewer #3:

*This is a very important study by Tupal, Huang et al. and Paul Gray as senior author*.

*This study has the potential to become an important conceptual and methodological contribution to the field, as it offers new ideas how inspiration and expiration are generated by the respiratory network. It is “generally” thought that inspiration and expiration are generated by two independent, yet interacting oscillators. One possibility is that the AtOH1 neurons generate expiration while the DBX1 neurons generate inspiration. Using elegant transgenic approaches the authors demonstrate that DBX1 neurons generate both respiratory phases and that the AtOH1 neurons establish the delay between these phases. This is a fundamental departure of the current concepts, which makes this study very interesting*. *The model proposed in this study is well explained. However, there are three main caveats that need to be addressed:*

*1) The activity characterized in this study represents an embryonic rhythm. While the authors argue that the activity is similar to a mature network activity, I am not convinced that the data presented are sufficiently strong to arrive at this conclusion*.

The question of whether the rhythmic output we describe is relevant for understanding adult behaviors is important. We respectfully disagree that what we recorded reflects an embryonic rhythm. Wild-type mice recovered by C-section at 18.5 are fully viable, breathe easily, and, if fostered, will survive to adulthood. These animals are not mature, but there is no evidence to suggest that their respiratory drives are generated by networks that become irrelevant one day later. As described above and in the modified paper, E18.5 mice generate a subset of breaths whose pattern of outputs are similar to what is seen in more mature in vitro and in vivo preparations. Moreover, Giles Fortin’s group showed very nicely that the early embryonic rhythm seen prior to e14.5 in mice is generated outside the ventral medulla in regions of the hindbrain that should be unaffected by either Atoh1 or Dbx1 mutation and that these rhythms vanish by E16.5 (95). Together we interpret these data to indicate the patterns of motor output we record are relevant for understanding mature respiratory networks.

*2) The identification of the respiratory phases is based on root recordings and not on actual EMG activity from inspiratory and expiratory muscles. This is a major comment, because the entire study hinges on the proper identification of inspiratory and expiratory activity*.

As described above, we have stepped back from using the terms inspiration and expiration to provide a more accurate description of the data. We have also analyzed both cervical and lumbar outputs in relation to either internal intercostal EMG recordings or VII root recordings and find that all three motor roots and EMG activity show similar patterns of independent function.

*3) The identification of a sigh burst is vague, which might lead to a misleading conclusion*.

As described above, we have significantly modified our description of biphasic sigh-like bursts in WT animals to better fit the data and changed our descriptions where they inaccurately described the literature. Most importantly, we performed considerable new experiments that now add additional examples of motor output from VII root and intercostal muscle activity

*These caveats could be addressed experimentally – by performing additional experiments, or by adding clearly stated caveats into the discussion*.

[Editors’ note: the author responses to the re-review follow.]

*1) The manuscript needs revision to make the presentation more accessible to a general audience. The data and figures and indeed the writing describing the data/figures are very complicated and the general reader will need a lot more guidance. Even experts in respiration need more guidance. Perhaps one method that will help the reader is to have the proposed circuit of*
Figure 8
*brought up front in*
Figure 1
*as an hypothesis to be tested and at the same time introduce the icons of*
Figure 8*. Then for each set of traces these icons could be used to guide the reader to see what interaction is being illustrated. The writing needs to use these iconic guides. The icons will also help is showing which neuron group is genetically eliminated in each experiment and/or is the target of peptide modulation or drugs and how these manipulations adjust the proposed circuit*.

*Creative use of icons and adjustments in writing could make the paper a lot easier to follow*.

We appreciate the advice to make the paper clearer for the general public and have made several significant changes to the text and figures. In the process of submission, it is sometimes extremely helpful to be asked to rethink how a paper is presented given the additional data acquired over time might be better used in a different manner.

First, we have modified the order of the paper to describe the details of the electrophysiological recordings from motor roots and EMG muscle to point out the basic experimental model and the putative relationship between specific hindbrain populations and the motor outputs from the different motor roots. To this we have also added modified versions of the putative network from Figure 8 to point out both features of motor output we will analyze (Figure 1). In this way we introduce the behaviors and hypotheses we are testing prior to describing the genetic manipulations and their effects.

Most importantly we have made significant modification to our figures to be clearer using a magenta rectangle to define the putative RTN/pFRG populations and a green circle to identify the preBötC. We have added a general schematic continuing the magenta rectangle and green circle describing (Figure 1) the candidate populations thought to be important for the generation of respiratory behaviors other than inspiration using the same color from Figure 1 for clarity. We have used this schematic to better explain the complicated genetics of the experiments throughout. This is clearest in Figure 3 where we show the motor patterns of cervical and lumbar output in WT and the four different transgenic mice models we used indicated by crossing out the populations targeted in each different mutation. Figure 3 repeats the schematic and description for clarity. In addition, we have modified the colors being used to identify specific genetic population to be different from those used to describe behavioral output for better consistency.

*The legends to the figures need to be made as clear as possible even if this means expanding the number of words, and they should refer to the icons in each case. We found the delay histograms of*
Figure 6
*and the phase histograms of*
Figure 7
*particularly difficult because they are described by overly dense and telegraphic legends. The supplemental figures need to be incorporated into the flow directly or eliminated*.

In general we have made our figure legends more descriptive and increasing their length by over 35%. For our descriptions of our analysis of the temporal lag (Figure 6) we have added a cartoon more clearly describing what we are measuring as well as the possible network interactions that might underlie those patterns similar to the initial description in Figure 1 and the summary hypothesis in Figure 8. For our description of phase analysis (Figure 7), we have similarly added descriptive model both of what we are measuring as well as the hypothesis being tested. In both cases we have modified the figure legends to better describe what is being measured and compared. We have also double-checked the description of the supplemental figures in the legends.

*Any way that the figures can be simplified by reducing the number of individual traces would be helpful*.

We have reduced the number of traces in Figure 4 and Figure 4—figure supplement 1 to limit the repetition and make the figures clearer. The rest of the traces provide essential descriptions of the behavioral output and are numerous because of the number of different transgenic mouse models tested or as a consequence of suggestions during the review process (e.g., Figure 6—figure supplement 1).

*2) The analysis of “sighs” remains a problem from the last submission. The authors should simply refer to these as double bursts in the Results and then in Discussion bring up their possible relationship to sighs. The double bursts should be quantified and a conservative course set in Discussion*.

We have modified the results to refer to these as biphasic cervical breaths so as to not confuse the reader and only mention briefly in the results their similarity to sighs to provide a context for their inclusion. Importantly, a just published paper from the Thoby–Brisson group (8) describes the onset of sigh production in the embryonic mouse and the necessity for inhibition for producing the biphasic breath but not the sigh itself. These results are completely consistent with our proposed respiratory model and our experimental results and we have included this paper in our interpretation in the Discussion. See also our response to the question 3 below.

*3) Another concern from the original submission is that only ∼25% of the normal bursts show 'normal' coordination. There should be some comment on this issue. How do you interpret this low percentage? It would seem to imply that the network capriciously teeters between*
Figure 8
*and*
Figure 8. *Is this how you see it?*

We have modified the discussion section to include newly published data describing changes in the strength of inhibitory synaptic drives in the respiratory network during late embryonic development as well as evidence for increased post-natal strengthening of inhibitory synapses. We now suggest less effective synaptic inhibition that strengthens post-natally could explain the differences we see in temporal delay at E18.5 as compared to older post-natal and adult animals. This fits both our model as well as the experimental results of other groups.

We interpret these results to indicate that the loss of RL neurons eliminates a net inhibitory effect on preBötC Dbx1 neurons, likely via reciprocally connected inhibitory interneurons. Recent work has found that biphasic inspiratory breaths (sighs) begin during late embryogenesis and that the temporal delay between eupneic and sigh components is dependent upon the strength of synaptic inhibition (8). Moreover, the density of the co-transporter necessary for setting the chloride equilibrium potential in the medulla increases significantly during the post-natal period (47).

This suggests the difference in the relative percentage of cervical-lumbar breaths showing temporal delays above 150ms between our E18.5 preparations and older animals may be simply a consequence of generally weaker synaptic inhibition (22). Importantly, we propose RL neurons play an essential role in the coordination of mammalian breathing behavior by modulating the inhibitory interneurons that produce the temporal lag between independent oscillators. This modulation also helps stabilize coupled oscillators preventing independent or unwanted activation. This has the advantage that temporal delays can be controlled without strongly affecting the rhythmogenic properties of the underlying oscillators (28; 23). It is also possible, however, that loss of Atoh1 neurons indirectly affects respiratory networks by modulating the development of inhibitory populations or the overall maturation of hindbrain circuits.

*4) The question of independent oscillators is complicated. It is possible to view the results as supporting 3 independent oscillators. On the other hand, as another reviewer points out, all the oscillations are driven by DBX1 cells and differential sensitivity to opiates has not been established so independent oscillators may not be supported. The authors need to address this issue squarely*.

This is an important issue and our data are only part of a much more complicated story. We have now added the following to our discussion to point out the limitations of our results.

It is important to point out that we were unable to selectively eliminate lumbar motor output in any of our genetic manipulations. While our data are consistent with the presence of independent respiratory oscillators, we cannot rule out that the variability we see between respiratory motor pools is the consequence of higher level network interactions between a single distributed respiratory oscillator and small pools of respiratory premotor neurons (85). Similarly, whether there are discrete anatomical boundaries between these putative independent oscillators is also unknown.